EMBO
Molecular Medicine

# The novel BET-CBP/p300 dual inhibitor NEO2734 is active in SPOP mutant and wild-type prostate cancer

Yuqian Yan[1,†], Jian Ma[1,2,3,†], Dejie Wang[1,†], Dong Lin[4,†], Xiaodong Pang[5], Shangqian Wang[6], Yu Zhao[1], Lei Shi[1], Hui Xue[4], Yunqian Pan[1], Jun Zhang[7], Claes Wahlestedt[8], Francis J Giles[9,‡], Yu Chen[6], Martin E Gleave[10], Collin C Collins[10], Dingwei Ye[2,3], Yuzhuo Wang[4,10,*] & Haojie Huang[1,11,12,**]

## Abstract

CULLIN3-based E3 ubiquitin ligase substrate-binding adaptor gene *SPOP* is frequently mutated in prostate cancer (PCa). PCa harboring SPOP hotspot mutants (e.g., F133V) are resistant to BET inhibitors because of aberrant elevation of BET proteins. Here, we identified a previously unrecognized mutation Q165P at the edge of SPOP MATH domain in primary and metastatic PCa of a patient. The Q165P mutation causes structural changes in the MATH domain and impairs SPOP dimerization and substrate degradation. Different from F133V hotspot mutant tumors, Q165P mutant patient-derived xenografts (PDXs) and organoids were modestly sensitive to the BET inhibitor JQ1. Accordingly, protein levels of AR, BRD4 and downstream effectors such as RAC1 and phosphorylated AKT were not robustly elevated in Q165P mutant cells as in F133V mutant cells. However, NEO2734, a novel dual inhibitor of BET and CBP/p300, is active in both hotspot mutant (F133V) and non-hotspot mutant (Q165P) PCa cells *in vitro* and *in vivo*. These data provide a strong rationale to clinically investigate the anti-cancer efficacy of NEO2734 in SPOP-mutated PCa patients.

**Keywords** BRD4; CBP/p300; NEO2734; prostate cancer; SPOP
**Subject Categories** Cancer; Pharmacology & Drug Discovery; Urogenital System

## Introduction

*SPOP* (speckle-type POZ protein) gene encodes a substrate recognition subunit of the CULLIN3-RBX1 E3 ubiquitin ligase (CRL) complex (Zhuang *et al*, 2009). SPOP binding generally triggers the ubiquitination and proteasomal degradation of target proteins mediated by RBX1-dependent recruitment of E2 ubiquitin-conjugating enzyme into the CRL complex. SPOP protein has several functional domains, including MATH (meprin and TRAF-C homology), BTB (Bric-a-brac/Tramtrack/Broad complex) and BACK (BTB and C-terminal Kelch; Zhuang *et al*, 2009). The MATH domain is responsible for substrate recruitment, while the BTB domain mediates dimerization and the BACK domain, either dimer or oligomer formation (Zhuang *et al*, 2009; Marzahn *et al*, 2016).

*SPOP* is the most frequently mutated gene in primary prostate cancer (PCa) and its mutation rate ranges from 10 to 15% of human PCa depending on the patient cohorts studied (Barbieri *et al*, 2012; Cancer Genome Atlas Research Network, 2015). Its mutations occur at a high frequency in a few specific residues (or so-called "hotspots") in the MATH domain, such as F133V, W131R, and F102C (Barbieri *et al*, 2012; Cancer Genome Atlas Research Network, 2015; Armenia *et al*, 2018). It seems that SPOP mutations are loss-of-function missense mutations which

1 Department of Biochemistry and Molecular Biology, Mayo Clinic College of Medicine and Science, Rochester, MN, USA
2 Department of Urology, Fudan University Shanghai Cancer Center, Shanghai, China
3 Department of Oncology, Shanghai Medical College, Fudan University, Shanghai, China
4 Department of Experimental Therapeutics, BC Cancer Research Centre, Vancouver, BC, Canada
5 Department of Physics, State Key Laboratory of Surface Physics, Fudan University, Shanghai, China
6 Human Oncology and Pathogenesis Program, Memorial Sloan Kettering Cancer Center, New York, NY, USA
7 Department of Laboratory Medicine and Pathology, Mayo Clinic College of Medicine and Science, Phoenix, AZ, USA
8 Center for Therapeutic Innovation, University of Miami Miller School of Medicine, Miami, FL, USA
9 Developmental Therapeutics Consortium, Chicago, IL, USA
10 The Vancouver Prostate Centre, Department of Urologic Sciences, University of British Columbia, Vancouver, BC, Canada
11 Department of Urology, Mayo Clinic College of Medicine and Science, Rochester, MN, USA
12 Mayo Clinic Cancer Center, Mayo Clinic College of Medicine and Science, Rochester, MN, USA
  *Corresponding author. Tel: +1-604-675-8013; E-mail: ywang@bccrc.ca
  **Corresponding author. Tel: +1-507-293-1311; E-mail: huang.haojie@mayo.edu
  †These authors contributed equally to this work
  ‡Formerly served as a consultant to Epigene Therapeutics Inc.

often result in deregulation and aberrant accumulation of its substrates (Barbieri *et al*, 2012; An *et al*, 2014; Zhang *et al*, 2017). Wild-type (WT) SPOP can self-assemble as higher-order oligomers and become a protein droplet to augment its substrate-binding function (Marzahn *et al*, 2016). It localizes to nuclear speckles while SPOP mutants exhibit a diffused pattern in the nucleus (Marzahn *et al*, 2016).

Bromodomain and extra-terminal domain (BET) family proteins BRD4 and others (BRD2, BRD3 and BRDT) function as epigenetic modifiers (chromatin readers) to facilitate gene transcription through context-specific interactions with acetylated histones and/or transcription factors. BET proteins can either elevate the expression of key oncogenic drivers such as c-MYC (Wyce *et al*, 2013) or enhance the activities of transcription factors such as androgen receptor (AR; Asangani *et al*, 2014) and ETS-related gene (ERG; Blee *et al*, 2016) in PCa. They can also activate the AKT signaling pathway via upregulation of RAC1 and several other genes (Zhang *et al*, 2017). To date, small molecule inhibitors specifically targeting the bromodomain of BET proteins are being tested in clinical or preclinical settings. However, BET inhibitors as a single therapeutic agent have exhibited limited activity (Zhang *et al*, 2017), and multiple mechanisms of resistance have been identified. We and others have previously reported that WT SPOP binds to and induces ubiquitination and proteasomal degradation of BET proteins by recognizing a common degron motif. PCa-associated SPOP hotspot mutants such as F133V and W131R impair both the binding and proteasomal degradation of BET proteins, leading to the resistance to BET inhibitors (Dai *et al*, 2017; Zhang *et al*, 2017).

Current PCa therapies targeting the androgen signaling axis focus on inhibition of AR or its ligand. Resistance mechanisms to these therapies usually involve the reestablishment of AR activity. In order to coordinate gene expression, AR acts together with numerous coactivator proteins including the chromatin writers such as histone acetyltransferases cAMP response element binding protein (CREB) binding protein (CBP) and p300 (Fu *et al*, 2000, 2003; Ianculescu *et al*, 2012). CBP/p300 proteins are key coactivators of AR and enhance the response to androgens. CBP/p300 proteins have an oncogenic role in PCa in a cellular context-dependent manner (Ding *et al*, 2014; Zhong *et al*, 2014), consistent with the finding that p300 is often deregulated in PCa patient samples (Debes *et al*, 2003). Tool compound small molecule CBP/p300 bromodomain inhibitors have demonstrated an important role for this domain in the coactivator functions of CBP/p300 and significant dose-dependent inhibition of AR signaling and PCa proliferation *in vitro* and *in vivo* (Comuzzi *et al*, 2004; Heemers *et al*, 2007; Lasko *et al*, 2017; Xiang *et al*, 2018).

Both AR and AKT signaling pathways are highly activated in SPOP mutant PCa (An *et al*, 2014; Geng *et al*, 2014; Blattner *et al*, 2017; Zhang *et al*, 2017). Targeting simultaneously these two pathways appears to be the key to treat SPOP mutant PCa (Yan *et al*, 2018). We investigated whether NEO2734, a novel small molecule inhibitor of both BET family and CBP/p300 bromodomains, might indirectly inhibit AKT signaling and AR activity. Our results demonstrated that this novel multi-epigenetic modifier inhibitor efficiently inhibited SPOP-mutated PCa cell growth in cell culture, organoid, PDX, and mouse xenograft models.

# Results

## Identification of heterozygous and homozygous SPOP Q165P mutations in PCa

To investigate novel genomic alterations during prostate tumorigenesis, exome sequencing was carried out in a cohort of 22 patients with PCa (CC Collins and YZ Wang, unpublished data). The sequencing analysis revealed a missense mutation in *SPOP* gene (Q165P, T > G) from primary PCa and liver metastasis biopsies in a patient (Figs 1A and B, and EV1A and B). This is a previously uncharacterized mutation with no overlap with those detected in a large cohort (> 1,000) of PCa patients (Cancer Genome Atlas Research Network, 2015; Armenia *et al*, 2018; Fig 1C). Intriguingly, the primary tumor in this patient contained a heterozygous Q165P mutation whereas the liver metastasis harbored a homozygous Q165P mutation (Fig 1A and B). To our knowledge, this is the first homozygous mutation of SPOP detected in PCa patients.

The MATH domain of SPOP contains 139 amino acids from amino acid 28 to 166. The Q165P mutated residue is located at the edge of the MATH domain and almost in the junction between the MATH and BTB domains (Zhuang *et al*, 2009; Fig 1C). The unique position of Q165P prompted us to determine how this mutation affects SPOP binding with substrates, SPOP dimerization, and the protein level of SPOP substrates.

The BET family proteins including BRD2, BRD3, and BRD4 have recently been identified as SPOP substrates and the elevation of these proteins due to hotspot mutations in SPOP confers BET inhibitor resistance in PCa (Dai *et al*, 2017; Zhang *et al*, 2017). We first examined the effect of the Q165P mutation on BRD2, BRD3, and BRD4 protein levels in the Q165P heterozygously mutated primary PCa biopsy using immunohistochemistry (IHC) (Figs 1D and EV1C). We also expanded BRD4 IHC staining in a large cohort (96 cases) of primary PCa patient specimens. The BRD4 protein level in Q165P mutant sample was comparable to that in other SPOP-mutated samples but much higher than that in the majority of SPOP WT samples (Fig 1D and E). Similarly, AR IHC staining in Q165P patient sample was comparable to that in the other SPOP-mutated samples and relatively higher than most of WT samples (Fig 1F and G).

Due to lack of sufficient tissue from the liver metastasis biopsy for IHC, we were unable to perform a similar analysis on the Q165P homozygously mutated patient sample but were able to perform in-depth studies (described below) in PDX tumors derived from the liver metastasis. Taken together, the IHC data suggest that Q165P mutation affects SPOP function in regulating the protein level of its substrates such as BRD4 and AR.

## Q165P impairs SPOP binding and ubiquitination of BRD4

The majority of SPOP mutations (~96%) detected thus far are missense heterozygous mutations that are located in the MATH domain (Cancer Genome Atlas Research Network, 2015; Armenia *et al*, 2018), a motif responsible for substrate recognition and interaction (Zhuang *et al*, 2009; Fig 1C). Similar to the findings reported previously (Zhang *et al*, 2017), SPOP hotspot mutation F133V almost completely abolished SPOP interaction with BRD4 (Fig 2A). In contrast, SPOP Q165P mutation only partially (~50%) diminished the ability of SPOP to bind to BRD4 (Fig 2A). We also compared the

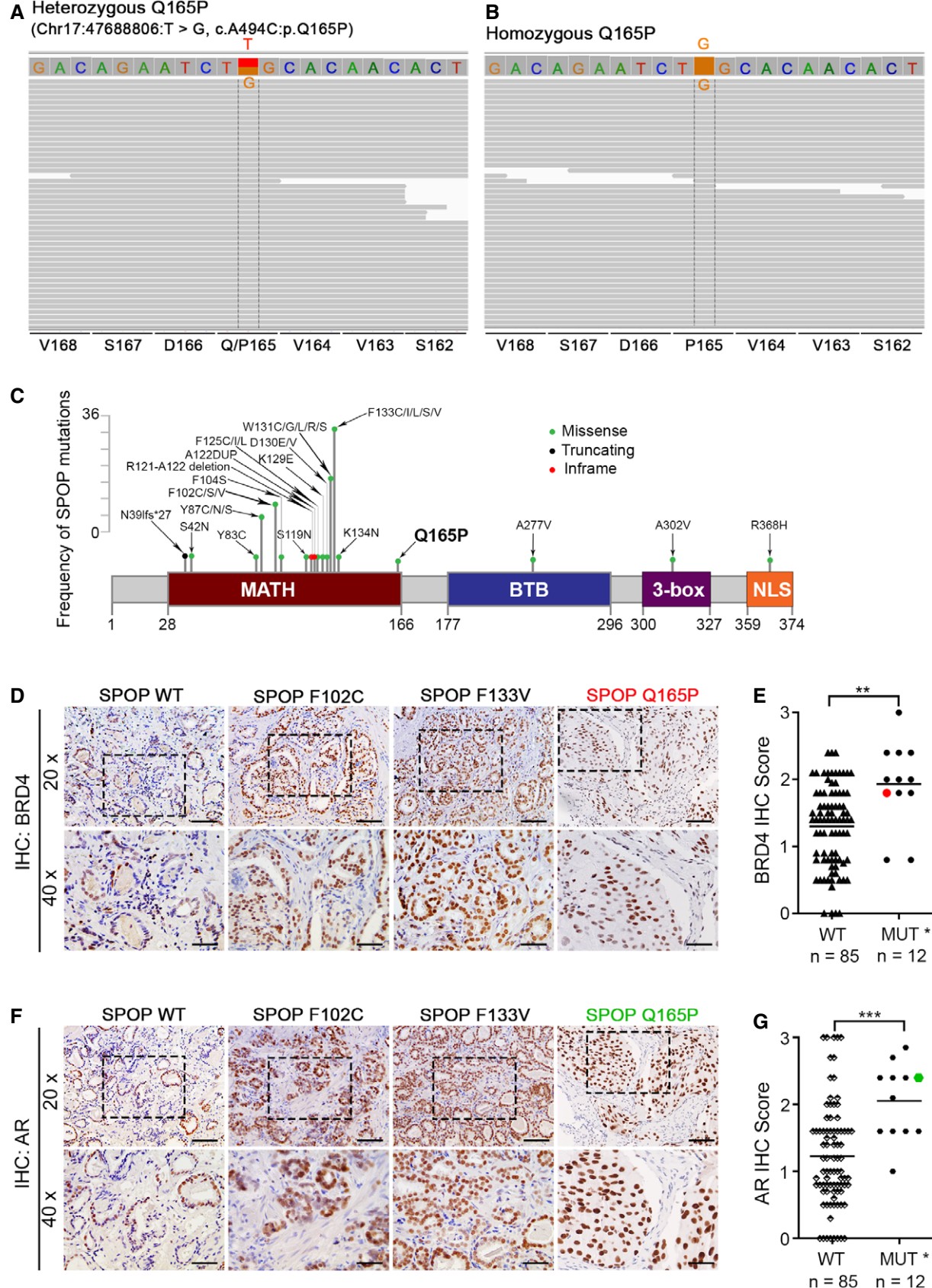

**Figure 1.**

◄

**Figure 1.  Identification of heterozygous and homozygous SPOP Q165P mutations and BET protein expression in PCa patient samples.**

A, B   A missense mutation in *SPOP* gene (Q165P, T > G) was identified in a PCa patient. Exome sequencing revealed that the primary tumor contained a heterozygous Q165P mutation (A) whereas the liver metastasis harbored a homozygous Q165P mutation (B).

C     A schematic shows a series of SPOP mutations including those detected in a large cohort of prostate adenocarcinomas [1,013 cases from MSKCC/DFCI (Armenia *et al*, 2018) and 499 cases from TCGA dataset (Cancer Genome Atlas Research Network, 2015)] and Q165P mutation identified in this report.

D, E   IHC analysis of BRD4 protein expression in SPOP wild-type (WT) and mutated (MUT) PCa patient samples. The representative images of BRD4 IHC staining in both SPOP WT and MUT PCa patients are shown in (D) and the quantified IHC data are shown in (E). Scale bars: 100 μm for 20× fields; 50 μm for 40× fields. The red dot in (E) indicates SPOP Q165P sample. The horizontal bar represents the mean. The *P* value was calculated by the unpaired two-tailed Student's *t*-test; **$P < 0.01$. See Appendix Table S4 for the detailed comparison, *P* values and sample number (*n*).

F, G   IHC analysis of AR protein expression in SPOP WT and mutated PCa patient samples. The representative images of AR IHC staining in both SPOP WT and mutant PCa patients are shown in (F) and the quantified IHC data are shown in (G). Scale bars: 100 μm for 20× fields; 50 μm for 40× fields. The green dot indicates SPOP Q165P sample. The horizontal bar represents the mean. The *P* value was calculated by the unpaired two-tailed Student's *t*-test; ***$P < 0.01$. See Appendix Table S4 for the detailed comparison, *P* values and sample number (*n*).

effect of F133V to that of Q165P on SPOP-mediated ubiquitination of BRD4 protein in LNCaP PCa cells, which expresses endogenous WT SPOP. Expression of Q165P substantially decreased SPOP-induced BRD4 polyubiquitination, but the effect of Q165P was much weaker than that of F133V (Fig 2B). Ectopic expression of Q165P in C4-2 cells increased the protein level of BRD4 and its downstream effectors such as RAC1 and phosphorylated AKT as well as other SPOP substrates such as AR and SRC3 (Zhang *et al*, 2017; Fig 2C). The effect of Q165P on most of the substrates was similar to that of F102C hotspot mutant but was less effective than F133V (Fig 2C). The differential effects of heterozygous expression of Q165P and F133V on BET protein level were pheno-copied in patient PDX samples (Fig 1D). These data indicate that the effect of Q165P mutation on SPOP functions, such as substrate binding and induction of substrate ubiquitination, is not as effective as some missense hotspot mutations when they are expressed in a heterozygous state.

Different from the effect of the heterozygous Q165P mutation, the protein levels of BRD4 and other substrates examined were much higher in Q165P homozygously mutated PDX tumors (573R) that established from the liver metastasis of the patient in comparison to SPOP WT PDX tumors (313HR), although expression of these proteins was still lower than that in F133V C4-2 xenograft tumors (Fig 2D). AR protein level in Q165P PDX tumors was slightly higher than the average AR protein level in castration-resistant PCa (CRPC) patient specimens (Fig EV1D and E). Consistent with the finding that both SPOP WT and Q165P PDX tumors were PTEN-null (Fig 2E), AKT was readily phosphorylated in both models (Fig 2D). Together, our data demonstrate that SPOP Q165P homozygous mutation shares the similar mechanism of action in substrate binding and degradation as SPOP hotspot mutations such as F133V and F102C, although the effect was not as robust as the hotspot mutants examined. Nevertheless, these data suggest that Q165P PDX represents a useful *in vivo* model for investigation of cancer biology and drug targeting of SPOP mutations in human PCa.

**Q165P mutation affects SPOP dimerization**

Dimerization is a prerequisite for SPOP recognition of substrates and its enzymatic activity in catalyzing substrate ubiquitination and degradation (Zhuang *et al*, 2009). Given the peripheral position of Q165 residue in the MATH domain (Fig 1C), the moderate impact of heterozygous Q165P mutation on SPOP binding with its substrates (Fig 2A) and substrate ubiquitination and degradation when it is expressed in a heterozygous state (Fig 2B and C), we

sought to determine whether Q165P mutation affects SPOP dimerization. Through molecular dynamics simulations, we observed that the average root-mean-square deviations (RMSDs) of atomic positions in WT and Q165P SPOP were 2.7 and 4.0 Å, respectively (Fig 3A). The greater RMSD of Q165P suggests that this mutation causes a larger conformational fluctuation and that the structure of Q165P is less stable than WT. The overall WT structure was very closed to the crystal structure, while the large deviation of Q165P structure from the crystal structure was observed from the simulation, especially in the MATH domain of Q165P (the green one) (Fig 3B). A close-up view showed that the Q165 residue is involved in the formation of the beta-sheet in the MATH domain (Fig 3C and D). In WT SPOP, the beta-sheet was stable during the simulation, and thus, the relative orientation of the two MATH domains was maintained. In contrast, the beta-sheet was partially broken in the P165-located area in the Q165P mutant (more obvious at the top right corner of Fig 3C and D), which exhibits the twisting of the MATH domain (green) of one molecule by approximately 35° compared to the WT. Overall, our simulation results indicate that Q165P mutation leads to a large conformational change in the MATH domain, which might subsequently affect the stability of SPOP dimer and ultimately impair SPOP-mediated ubiquitination of its substrates. These analyses also reveal that there are several amino acids important for dimerization. Mutations on these sites might destroy the dimer interface for MATH domain. Specifically, the conversion of any residue of amino acids 161–165 to proline at the region where Q165 locates or conversion of any residue of amino acids 27–32 to proline at the area on the dimer interface (Fig 3E).

To experimentally verify whether Q165P mutation affects SPOP dimerization, both WT SPOP and Q165P mutant were expressed with two different protein tags. Co-immunoprecipitation (co-IP) assay showed that SPOP dimerization was substantially impaired when Q165P mutant was co-expressed with WT SPOP, but the effect was much enhanced when Q165P mutant was expressed in the homozygous manner (Fig 3F). These data are also consistent with the differential impact of homozygous and heterozygous Q165P mutation on the elevation of BRD4 protein (Fig 2C and D), although Q165P heterozygously mutated cells were cultured *in vitro* (Fig 2C) and homozygous tumors grew in mice (Fig 2D). Furthermore, co-IP assays showed that CULLIN3, the partner of SPOP required for the E3 ubiquitin ligase activity of the complex, interacted much weaker with Q165P than SPOP WT (Fig 3G). Thus, our data suggest that Q165P affects several aspects of SPOP functions, including substrate binding, dimerization, and enzymatic activity.

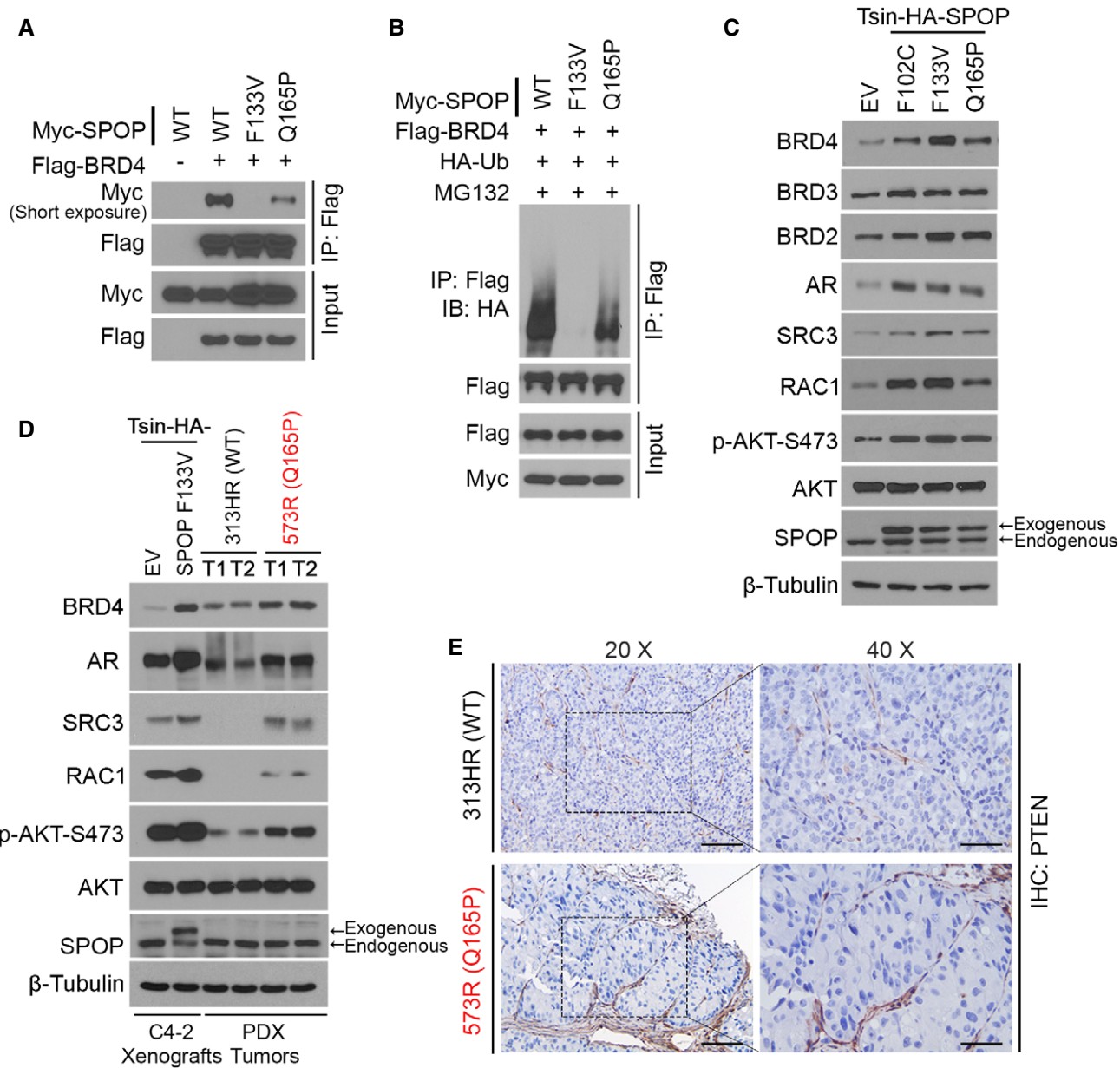

**Figure 2. Q165P mutation partially impairs SPOP interaction with BRD4 and undermines BRD4 ubiquitination.**

A, B    LNCaP cells were transfected with the indicated plasmids for 48 h and harvested for Western blot analysis with the indicated antibodies to detect the interaction between BRD4 and SPOP (A) and BRD4 ubiquitination (B).

C    C4-2 cells were stably infected with the lentivirus expressing empty vector (EV) or SPOP mutants and harvested for Western blot analysis with the indicated antibodies.

D    C4-2 xenograft tumors expressing EV or F133V and SPOP WT and Q165P PDX tumors were harvested for Western blot analysis with the indicated antibodies. T1: tumor 1; T2: tumor 2.

E    Representative IHC images of PTEN protein in SPOP WT and Q165P PDX tumors. Scale bars: 100 μm for 20× fields; 50 μm for 40× fields.

Source data are available online for this figure.

## Q165P mutant PCa cells are sensitive to both JQ1 and NEO2734

It has been shown previously that SPOP hotspot mutations such as F133V and W131R confer JQ1 resistance in PCa cells by stabilizing BET proteins (Dai *et al*, 2017; Zhang *et al*, 2017). Given that the effect of Q165P on the elevation of BET proteins is not as robust as F133V, we sought to determine the sensitivity of Q165P cells to JQ1. To this end, we established Q165P stable DU145 cells. Western blot analysis showed that the levels of Q165P and endogenous WT SPOP were comparable (Fig 4A), closely mimicking the heterozygous status of SPOP mutation in PCa patient samples. Similar to the effect of F133V as reported previously (Zhang *et al*, 2017), the cell

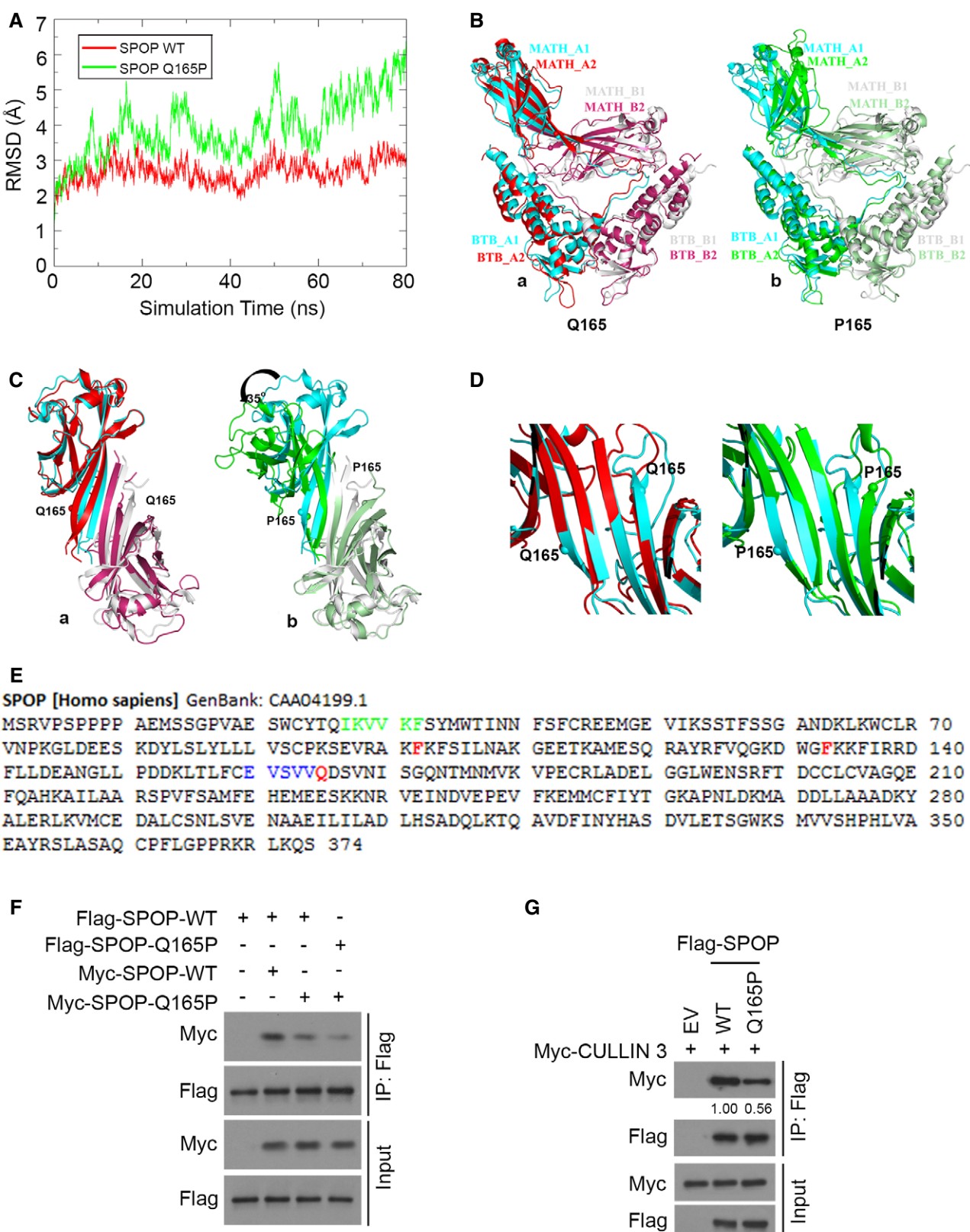

**Figure 3.**

**Figure 3. Q165P mutation affects SPOP dimerization.**

A   Root-mean-square deviation (RMSD) calculations of both SPOP WT and Q165P mutant on CA atoms with reference to the crystal structure (3HQI.pdb).

B   Overall architecture of SPOP dimer. Superposition of all Cα atoms of WT structure (a) and Q165P structure (b) from simulation at 80 ns over the crystal structure. The two SPOP molecules in the crystal structure are colored cyan and gray, respectively; the WT structure from simulation are red and warm pink, and Q165P are green and pale green, respectively.

C, D   Close-up views of the MATH domain from top of (B). The Cα atoms of Q165 or P165 are represented by spheres.

E   Several amino acids are predictively important for SPOP dimerization. Mutations on the following residues might destroy the dimer interface for MATH domain: any residue in blue color to Pro at the strand where Q165 locates or any residue in green color to Pro at the strand on the dimer interface. Residues mutated in PCa patients F102, F133, and Q165 are in red.

F, G   Co-IP assay was performed to detect the efficiency of dimerization between SPOP WT and Q165P mutant (F) and the interaction between CULLIN3 with SPOP WT or Q165P (G).

Source data are available online for this figure.

viability assay showed that Q165P mutant DU145 cells grew faster than empty vector (EV) control cells (Fig 4B). Intriguingly, Q165P DU145 cells remained sensitive to JQ1 although the treatment in Q165P cells was less effective than in the control cells (Fig 4B). Both control and Q165P DU145 cells were sensitive to a tool compound CBP/p300 inhibitor CPI-637 (Fig 4B). Since NEO2734, a novel BET and CBP/p300 dual inhibitor, offers a unique opportunity to explore the potential inhibition of both BET and CBP/p300 pathways with a single agent (Giles *et al*, 2018), we explored its activity in Q165P mutant PCa cells. NEO2734 caused significant inhibition of cell growth, even superior to the effect of the combination of JQ1 and CPI-637 (Fig 4B). Notably, Q165P mutant cells were more sensitive to NEO2734 than the control cells (Fig 4B). Moreover, clonogenic survival assay showed that the IC50 of NEO2734 was lower in Q165P cells than EV cells (0.69 μM versus 1.08 μM) (Fig 4C). When treated with NEO2734 at a concentration (1 μM) close to IC50 in EV cells, Q165P cells grew fewer colonies than EV cells (Fig 4D and E), confirming that Q165P mutant cells are more sensitive to NEO2734 than the control cells.

Given that the dual inhibitor NEO273 has shown a much superior inhibition of cell growth than co-targeting both pathways individually with JQ1 and CPI-637, we sought to investigate the underlying mechanism(s). We examined BRD4 occupancy and histone H3 acetylation (Ac-H3) level in BRD4 known target genes *c-MYC* and *RAC1* (Zhang *et al*, 2017) in DU145 cells using chromatin immunoprecipitation quantitative PCR (ChIP-qPCR). While NEO2734 and dual administration of JQ1 and CPI-637 achieved a similar effect on BRD4 binding at *c-MYC* and *RAC1* gene loci (Fig 4F), NEO2734 achieved much greater inhibition on Ac-H3 level at these two gene loci than the dual inhibition by JQ1 and CPI-637 (Fig 4G). Similar results were obtained in C4-2 cells (Fig EV2A and B). These data suggest that in comparison with the combined treatment of CBP/p300 and BET inhibitors, the greater inhibition achieved by the dual inhibitor NEO2734 may be attributed, at least in part, to its greater inhibition of Ac-H3 level at BRD4 target gene loci.

**Q165P mutant PDX tumors are sensitive to both JQ1 and NEO2734**

More and more PDX models are being utilized for the purpose of precision medicine. They also allow researchers to access more clinically relevant *in vivo* models to investigate the etiology of different cancer subtypes. To further investigate the functional relevance of SPOP Q165P mutation *in vivo*, we generated a PDX model from the Q165P homozygous mutant liver metastatic biopsy (Fig 5A). To our

knowledge, this is the first report of SPOP homozygous mutant PCa PDX model. Sanger sequencing confirmed that PDX tumors harbored homozygous SPOP Q165P mutation as the origin biopsy (Fig EV3A). In agreement with the finding in the biopsy of the patient (Figs 1D and EV1C), Western blot analysis demonstrated that BRD2, BRD3 and BRD4 protein expression was higher in Q165P mutant PDX line compared to two different SPOP WT PDX lines (Fig EV3B and C). IFC showed that both AR and phosphorylated AKT proteins were also elevated in Q165P PDX tumors compared to control PDXs (Fig EV3D). These data provide evidence that Q165P mutation can enhance both BET and AR signaling pathways *in vivo*. Moreover, we performed E-cadherin IFC to define the cell boundary in Q165P PDX samples. Intriguingly, we noticed that there were a few cell clusters with stronger E-cadherin IFC signal in Q165P PDX samples (Fig EV3D). However, quantitative analysis of E-cadherin IFC signal indicated that no significant difference in the fluorescence intensity was detected between SPOP WT and Q165P mutant PDX tumors (Fig EV4A and B). This observation was further confirmed by Western blot analysis (Fig EV4C and D). While low expression of E-cadherin has been linked to epithelial–mesenchymal transition (EMT; Kalluri & Weinberg, 2009), our data suggest that Q165P mutation less likely has an obvious effect on EMT in the PDX tumors.

To test the anti-cancer efficacy of NEO2734 in the Q165P PDX model, we expanded PDX tumors in severe combined immunodeficiency (SCID) mice for drug treatment (Fig 5A). We treated SPOP WT and Q165P PDX tumors with NEO2734, JQ1, or CPI-637 alone or both. Similar to the finding in Q165P mutant DU145 cells in culture (Fig 4B), Q165P mutant PDX tumors were also responsive to JQ1 although the effect was not drastic as the combined treatment with JQ1 and CPI-637 or NEO2734 alone (Fig 5B and C). While both AKT and AR signaling pathways were activated in SPOP mutant PDXs (Figs 2D and 5D), JQ1 treatment alone inhibited Q165P mutation-induced increase in AKT phosphorylation and expression of AR and its downstream target genes such as *PSA* (*KLK3*), *TMPRSS2*, and *NKX3.1;* the effect of NEO2734 or co-treatment of JQ1 and CPI-637 was much more robust (Fig 5D and E). Thus, the dual inhibitor NEO2734 can inhibit the activation of both AKT and AR signaling in Q165P mutated PCa PDX tumors, resulting in the suppression of tumor growth *in vivo*.

**NEO2734 is active in JQ1-resistant SPOP hotspot mutant PCa organoids**

We demonstrated previously that organoids expressing SPOP W131R, a hotspot mutation, are resistant to JQ1 (Zhang *et al*,

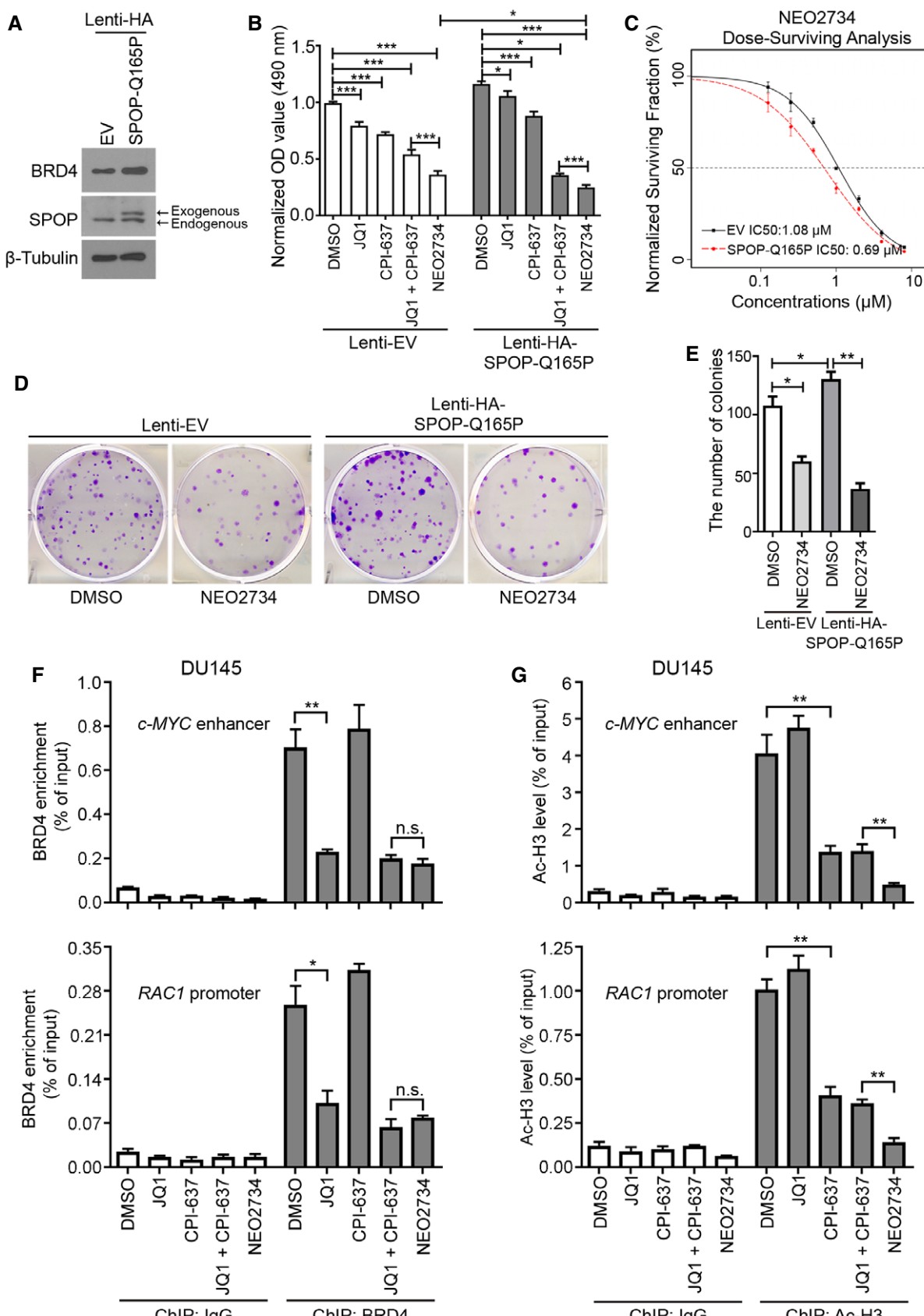

**Figure 4.**

**Figure 4.  BET and CBP/p300 dual inhibitor NEO2734 overcomes JQ1 resistance in SPOP-mutated PCa cells *in vitro*.**

A  DU145 PCa cells infected with virus expressing EV or SPOP Q165P were harvested for Western blot analysis.

B  MTS was performed to compare the growth between EV and Q165P DU145 cells after the treatment with indicated inhibitors (2 μM of JQ1, 2 μM of CPI-637, 2 μM NEO2734) for 72 h. All data shown are means ± SEM, $n$ = 15 (DMSO group) and $n$ = 5 (all other groups). The $P$ value was calculated by the unpaired two-tailed Student's $t$-test; *$P$ < 0.05, ***$P$ < 0.001. See Appendix Table S4 for the detailed comparison, $P$ values and sample number ($n$).

C–E  Clonogenic survival assay was performed to determine the sensitivity of NEO2734 in SPOP Q165P DU145 cells. The survival curve showed IC50 for SPOP Q165P (0.69 μM) and EV cells (1.08 μM) (C). DU145 cells expressing lenti-EV or lenti-HA-SPOP-Q165 constructs were treated with 1 μM of NEO2734 for 4 days and cultured for another 8 days before harvest. The number of colonies with more than 50 cells was counted from three replicates ($n$ = 3). Representative images of colonies are shown in (D) with quantitative data shown in (E). All data shown are means ± SEM, $n$ = 4 (panel E). The $P$ value was calculated by the unpaired two-tailed Student's $t$-test; *$P$ < 0.05, **$P$ < 0.01. See Appendix Table S4 for the detailed comparison, $P$ values and sample number ($n$).

F  DU145 cells were treated with the indicated inhibitors and harvested for ChIP with BRD4 antibody. The enrichment of BRD4 at *c-MYC* enhancer and *RAC1* promoter was analyzed using qPCR. All data shown are means ± SEM, $n$ = 3. The $P$ value was calculated by the unpaired two-tailed Student's $t$-test; n.s., not significant, *$P$ < 0.05, **$P$ < 0.01. See Appendix Table S4 for the detailed comparison, $P$ values and sample number ($n$).

G  DU145 cells were treated with the indicated inhibitors and harvested for ChIP with Ac-H3 antibody. The level of Ac-H3 at *c-MYC* enhancer and *RAC1* promoter is analyzed using qPCR. All data shown are means ± SEM, $n$ = 3. The $P$ value was calculated by the unpaired two-tailed Student's $t$-test; **$P$ < 0.01. See Appendix Table S4 for the detailed comparison, $P$ values and sample number ($n$).

Source data are available online for this figure.

2017). Next, we sought to employ clinically relevant organoid models to determine whether SPOP hotspot and non-hotspot mutant PCa cells respond differently to JQ1 and NEO2734. To this end, we first established organoids from SPOP WT (control) and Q165P mutant PDXs using the method as described previously (Drost *et al*, 2016; Fig EV5A). Sanger sequencing confirmed that the organoids retained the Q165P mutant status after culturing *in vitro* (Fig EV5B). We demonstrated that after 10-day culture *in vitro*, WT organoids tended to form a hollow and round-shaped structure and some of them were even collapsed. In contrast, Q165P mutant organoids formed solid and irregular spheres and adhered to the bottom of culture dishes (Fig EV5C). Quantification of the diameter of organoids showed that Q165P organoids were bigger than WT counterparts (Fig EV5D). These data suggest that the homozygous SPOP Q165P mutation confers a more aggressive cancer phenotype, which is consistent with the metastatic origin of this PDX in the patient.

To investigate whether our findings are generalizable to other SPOP mutant organoids, we examined the anti-cancer effect of JQ1 and CPI-637 co-treatment or NEO2734 alone using multiple PCa organoid lines. We compared the alteration of AR and AKT signaling pathways among six organoids, two harboring SPOP mutation (Q165P or W131R), and other four expressing SPOP WT (Gao *et al*, 2014; Zhang *et al*, 2017). In agreement with the findings in PDX (Fig 5D and E) and previous studies in mice (Blattner *et al*, 2017), AR and phosphorylated AKT levels were much higher in SPOP mutant organoids than SPOP WT counterparts (with an exception of AKT phosphorylation in one SPOP WT organoid and an exception of AR expression in another SPOP WT organoid; Figs 6A and B, and EV5E).

Since both AKT and AR signaling pathways are activated in SPOP-mutated PCa organoids, we examined whether NEO2734 can efficiently inhibit their growth. While all four SPOP WT organoids were responsible to all types of treatment, Q165P organoids were sensitive, but the W131R hotspot mutant organoids (ASC1) were resistant to JQ1 (Fig 6C and D). Notably, both Q165P and W131R mutant organoids were extremely sensitive to the combined treatment with JQ1 and CPI-637 or NEO2734 alone (Fig 6C and D).

To determine the molecular mechanisms underlying the differential responsiveness of SPOP WT and mutant organoids to various drug treatments, we examined both apoptotic marker cleaved caspase-3 and proliferation marker Ki67 in organoids. JQ1 caused a significant increase of caspase-3-positive cells in both WT and Q165P mutant organoids (Fig 6E and F). The combination of JQ1 and CPI-637 or NEO2734 significantly increased the number of apoptotic cells in both SPOP WT and mutant organoids (Fig 6E and F). Interestingly, these compounds had no effects on the cell proliferation as indicated by little or no significant alteration of proliferation marker Ki67 in all treatment groups of both SPOP WT and Q165P organoids (Fig EV5F and G). Intriguingly, consistent with increased growth of Q165P mutant DU145 cells in culture (Fig 4B), Q165P mutant organoids displayed more Ki67-positive staining compared with SPOP WT counterparts (Fig EV5F and G). Together, our data demonstrate that both SPOP hotspot (W131R) and non-hotspot mutant (Q165P) organoids are sensitive to the dual inhibitor of BET and CBP/p300 bromodomains.

**NEO2734 is active in JQ1-resistant SPOP hotspot mutant PCa xenografts in mice**

The findings in organoids prompted us to determine whether JQ1-resistant SPOP hotspot mutant PCa xenografts are sensitive to NEO2734 *in vivo*. Similar to our previous report (Zhang *et al*, 2017) and the results from organoid studies (Fig 6), cell viability assays showed that C4-2 cells expressing the SPOP hotspot mutant F133V were resistant to JQ1 while Q165P C4-2 cells were modestly responsive to JQ1 (Fig 7A and B). Importantly, both F133V and Q165P cells were robustly responsive to NEO2734 (Fig 7B). This observation was further supported by the results from clonogenic survival assay in C4-2 cells (Fig 7C–E). Similar to the findings in the PDX model (Fig 5) and our previous report (Zhang *et al*, 2017), Q165P xenografts were modestly responsive whereas F133V tumors were resistant to JQ1 treatment (Fig 7F–H). In agreement with the observation that the elevated BRD4 level and the downstream signaling are critical for JQ1 resistance in PCa cells harboring SPOP hotspot mutants such as F133V and W131R (Zhang *et al*, 2017), the protein levels of BRD4 and phosphorylated AKT were lower in Q165P mutant cells compared to F133V expressing cells, organoids, and PDX tumors (Figs 2C and D and 6A). Most importantly, NEO2734 induced marked suppression of growth of both Q165P and F133V C4-2 xenografts in

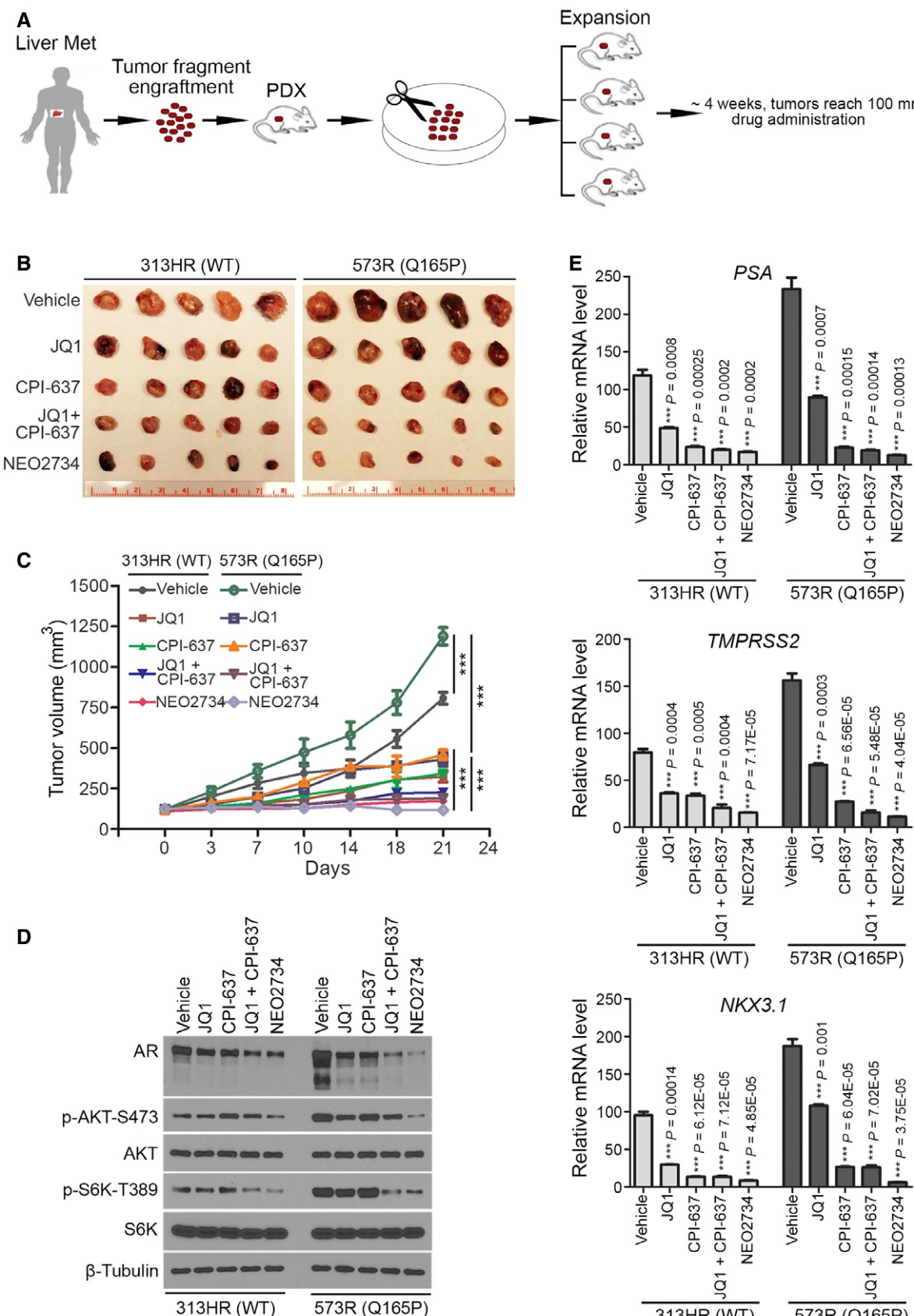

**Figure 5.**

◀

**Figure 5. Q165P mutant PDX tumors are sensitive to the BET and CBP/p300 dual inhibitor NEO2734.**

A  A schematic depicts the procedure of the establishment of PDX and drug administration.

B, C  SCID male mice with PDX tumors were treated with vehicle (40% polyethylene glycol), JQ1 (50 mg/kg), CPI-637 (30 mg/kg), combination of JQ1 (50 mg/kg) and CPI-637 (30 mg/kg) or NEO2734 (30 mg/kg) five days a week for three consecutive weeks. Tumors isolated from mice at day 21 of drug treatment were photographed (B) and tumor growth is shown in (C). All data shown are means ± SEM, $n = 10$ except for Q165P (CPI-637, $n = 9$) and Q165P (NEO2734, $n = 7$). The $P$ value comparing the tumor volume at day 21 post-treatment was calculated by the unpaired two-tailed Student's $t$-test; ***$P < 0.001$. See Appendix Table S4 for the detailed comparison, $P$ values and sample number ($n$).

D, E  PDX tumors in each treatment group were harvested for protein and RNA extraction. For Western blot analysis, three tumors each group ($n = 3$) were used for protein extraction and mixed with equal ration to detect the indicated antibodies in (D). For RT–qPCR, three tumors each group ($n = 3$) were extracted RNA for RT–qPCR in (E). Data are shown as means ± SEM. The $P$ value comparing vehicle with each treatment group was calculated by unpaired two-tailed Student's $t$-test and indicated in the graph.

Source data are available online for this figure.

mice (Fig 7G and H), suggesting that the dual inhibitor works effectively in both SPOP hotspot and non-hotspot mutant tumors *in vivo*.

## Discussion

*SPOP* is the most frequently mutated gene in primary PCa (Barbieri *et al*, 2012; Cancer Genome Atlas Research Network, 2015; Armenia *et al*, 2018). The SPOP-mutated subtype of PCa possesses many unique features including mutual exclusivity with *TMPRSS2-ERG* gene fusions, co-occurrence with *CHD1* gene deletion, and genomic instability, thereby constituting a unique subset of disease. Increasing evidence indicates that SPOP mutations often cause elevation of its degradation substrates including AR and transcriptional cofactors such as TRIM24, SRC3, and BET proteins (An *et al*, 2014; Geng *et al*, 2014; Groner *et al*, 2016; Blattner *et al*, 2017; Zhang *et al*, 2017). Consistent with the finding that SPOP-mutated specimens acquire highest AR activity among different subtypes of PCa (Cancer Genome Atlas Research Network, 2015), SPOP-mutated PCa cells are highly sensitive to treatment of androgen inhibitory agent abiraterone when *SPOP* mutation is co-occurred with *CHD1* deletion (Boysen *et al*, 2018), supporting the notion of oncogene addiction. In contrast, PCa-associated SPOP hotspot mutations such as F133V and W131R confer resistance to BET inhibitors due to upregulation of BET proteins and aberrant occupancy of BRD4 in the genome (Dai *et al*, 2017; Zhang *et al*, 2017), implying that specific strategies are needed to effectively treat patients with SPOP-mutated PCa. In agreement with the previous finding that treatment of SPOP-mutated PCa cells with either the BET inhibitor JQ1 or the CBP/p300 inhibitor C646 invariably diminishes BRD4 enrichment at its binding loci (Zhang *et al*, 2017), we demonstrated in the current study that SPOP mutant PCa cell lines, organoids, and PDX tumors were sensitive to co-treatment with JQ1 and the CBP/p300 inhibitor CPI-637. Most importantly, simultaneous inhibition of the BET family and CBP/p300 proteins with the dual pathway inhibitor NEO2734 results in equivalent or superior effect in comparison with that achieved by co-targeting both pathways with individual inhibitors. These findings provide a strong rationale for the inclusion of patients with advanced SPOP-mutated PCa in clinical studies of NEO2734.

*In vivo* PDXs and *in vitro* organoids of PCa are valuable preclinical models that faithfully recapitulate the genomic complexity and phenotypic diversity of this malignancy. To date, however, only a handful of SPOP mutant PCa PDX and organoid models have been reported. A missense mutation (Y83C, A > G) was detected via exome sequencing in LuCaP147 PDX (Kumar *et al*, 2011). An *in vitro* LuCaP147 spheroid culture was further derived from this PDX and most importantly, SPOP Y83C mutation was faithfully inherited in this spheroid culture (Saar *et al*, 2014). Another valuable feature of LuCaP147 culture and PDX models is that both of them are PTEN-positive (Saar *et al*, 2014), mimicking the situation in patients that SPOP mutations commonly occurred in PTEN-positive PCa (Cancer Genome Atlas Research Network, 2015; Armenia *et al*, 2018). An interesting feature of the LuCaP147 model is that it has a hypermutated phenotype (Kumar *et al*, 2011), an observation consistent with the discovery that SPOP mutant cells are genomically unstable (Boysen *et al*, 2015). At least two SPOP mutant PCa organoids have been reported previously (Gao *et al*, 2014; Zhang *et al*, 2017), one harbors heterozygous F133L and the other expresses heterozygous W131R mutation, two hotspot mutations in the MATH domain of SPOP (Fig 1C). In the current study, we for the first time report SPOP Q165P mutation in PCa patient biopsies, PDX tumors, and organoids. Most importantly, we demonstrated that similar to other SPOP mutants, Q165P mutation undermined the ability of SPOP to degrade its substrates such as AR and BRD4. It is worth noting that Q165P mutated tumors and PDX are PTEN-negative. While it has been shown that SPOP mutations are generally mutually exclusive with PTEN deletion or mutation in early, clinically localized PCa, these two lesions do co-exist in some advanced PCa in patients (Haffner *et al*, 2013; Robinson *et al*, 2015). Indeed, prostate-specific expression of SPOP F133V in combination with *Pten* homozygous deletion further enhances PCa progression in mice (Blattner *et al*, 2017). Thus, the discovery of this new mutation and establishment of Q165P mutant PDX and organoid models expand our capacity to investigate the biological and therapeutic significance of SPOP mutations *in vitro* and *in vivo*.

Unlike hotspot mutations of SPOP, while Q165P occurred as a heterozygous mutation in primary PCa, it was a homozygous mutation detected in liver metastasis in the same patient. The findings from the current study and those reported previously provide several mechanistic explanations for the occurrence of Q165P homozygous mutation. Q165P mutation occurs at the peripheral position in the MATH domain, a motif required for substrate binding. This observation is consistent with the finding that compared to the hotspot mutation F133V, heterozygous Q165P only partially impaired the ability of SPOP to bind and degrade BRD4 protein (Fig 2A and B). Thus, it is conceivable that homozygous mutation at the Q165 residue might have been

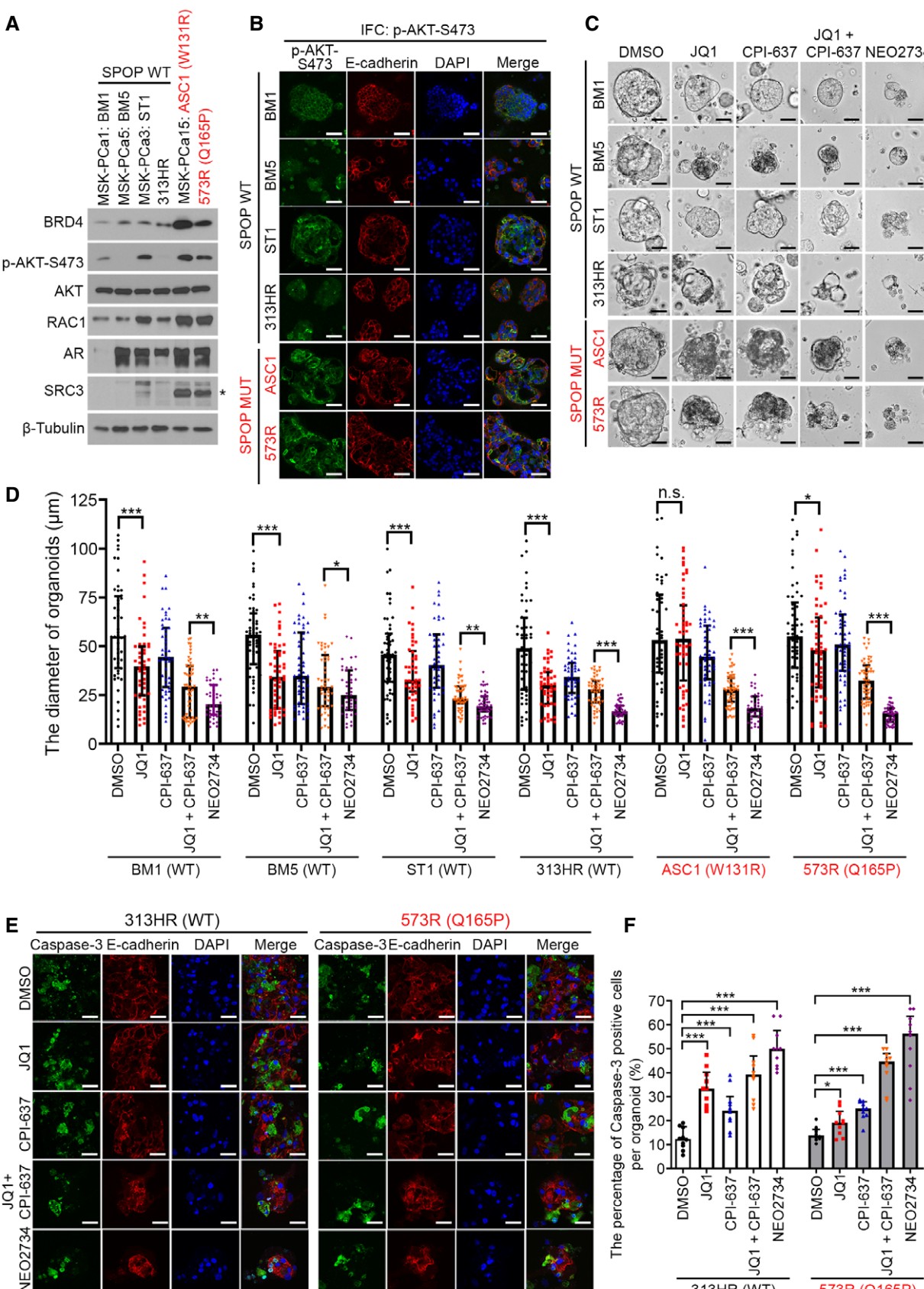

Figure 6.

**Figure 6. JQ1-resistant SPOP hotspot mutant organoids are sensitive to NEO2734.**

A, B  Six organoid lines including four SPOP WT (BM1, BM5, ST1, and 313HR) and two MUT (573R and ASC1) were harvested for Western blot analysis with the indicated antibodies (A) and IFC with p-AKT-S473 antibody (B). Asterisk, SRC3 at expected molecular mass. E-cadherin antibody was used to indicate the cell membrane (red) and DAPI for nucleus (blue). Scale bars: 25 μm.

C, D  Six organoid lines including four SPOP WT (BM1, BM5, ST1, and 313HR) and two MUT (573R and ASC1) were cultured for 5 days, followed by the treatment with JQ1 (2 μM), CPI-637 (2 μM), JQ1 (2 μM) + CPI-637 (2 μM) or NEO2734 (2 μM) for five more days. The representative images of organoids after the treatment are shown in (C) and the quantified data of the organoid diameter are shown in (D). Scale bars: 25 μm. All data shown are means ± SEM, $n \geq 38$. The P value was calculated by the unpaired two-tailed Student's t-test; n.s., not significant, *$P < 0.05$, **$P < 0.01$, ***$P < 0.001$. See Appendix Table S4 for the detailed comparison, P values and sample number (n).

E, F  313HR (SPOP WT) and 573R (Q165P) organoids were cultured for 5 days, followed by the treatment with the indicated inhibitors for 5 days. The organoids were stained with caspase-3 antibody (green) to detect apoptotic cells. E-cadherin antibody staining was used to define the cell membrane (red) and the nucleus was counterstained with DAPI (blue). The representative images of cleaved caspase-3 IFC staining are shown in (E). Scale bars: 25 μm. All data shown are means ± SEM, $n = 10$. The P value was calculated by the unpaired two-tailed Student's t-test; *$P < 0.05$, ***$P < 0.001$. See Appendix Table S4 for the detailed comparison, P values, and sample number (n).

Source data are available online for this figure.

selected due to its stronger causal effect on SPOP substrate stabilization and PCa oncogenesis and progression. This notion is fully supported by our finding that the level of the SPOP substrate BRD4 was elevated in a much greater degree in Q165P homozygous tumors compared to Q165P heterozygous cells. Conversely, for hotspot mutants such as F133V that reside in the middle of MATH domain and happen to be on the substrate-binding surface (Barbieri *et al*, 2012), a heterozygous mutation has already drastically affected SPOP protein functions. Therefore, it is not surprising that no second hit has been reported for hotspot mutants in the MATH domain (Armenia *et al*, 2018).

Our structure modeling studies also provide further molecular insight into the functional differences between Q165P and SPOP hotspot mutations. The computation simulation performed in the current study showed that dimerization of Q165P mutant is much different from the WT counterpart. In contrast, our previous computer-based modeling showed that in the case of ERG as the substrate, the binding of hotspot mutant F133V with ERG is apparently impaired (An *et al*, 2015). Our co-IP studies confirmed that SPOP binding with its substrates was completely abolished by F133V. In contrast, while substrate binding of SPOP was only partially compromised by Q165P, this mutation does impair SPOP dimerization. Thus, our findings not only reveal the functional differences between Q165P and the hotspot mutations, but also provide molecular insights into their differences.

In summary, we identified for the first time the SPOP homozygous mutant Q165P and generated PDX and organoid models from this unique mutant. Given that only a handful of SPOP-mutated cell line, organoid and PDX models have been generated and are available in the field, the discovery of this new mutation and establishment of the Q165P mutant PDXs and organoids expand our capacity to investigate the biological and therapeutic significance of the SPOP-mutated subtype of PCa. Consistent with the evidence that this mutation occurs in the very edge of the substrate-binding MATH domain, Q165P mutation only partially impairs SPOP binding with its substrates such as BRD4. Further mechanistic studies reveal that Q165P mutation, especially in the homozygous status, also largely diminishes SPOP functions in dimerization and ubiquitination-catalyzing activity, thereby providing a mechanistic explanation as to why the level of BRD4 protein was substantially elevated in Q165P homozygous organoids and PDX tumors. While SPOP hotspot mutants confer BET inhibitor resistance, Q165P mutant cells are modestly sensitive to the BET inhibitor JQ1, and this finding is consistent with the moderate increase in BET proteins and its downstream signaling molecules in Q165P mutant PCa organoids and PDX tumors. Given that both SPOP hotspot and non-hotspot mutant cells are hypersensitive to the BET and CBP/p300 dual inhibitor, NEO2734, these findings provide a strong rationale for the inclusion of patients with SPOP-mutated PCa in clinical studies of NEO2734.

**Figure 7. JQ1-resistant SPOP hotspot mutant xenografts are sensitive to NEO2734.**

A  C4-2 PCa cells infected with virus expressing EV, SPOP Q165P or F133V were harvested for Western blot analysis.

B  MTS was performed to compare the cell growth between EV, Q165P, F133V C4-2 groups after the treatment with indicated inhibitors (2 μM of JQ1, 2 μM of CPI-637, 2 μM NEO2734) for 72 h. All data shown are means ± SEM, $n = 7$. The P value was calculated by the unpaired two-tailed Student's t-test; n.s., not significant, *$P < 0.05$, **$P < 0.01$, ***$P < 0.001$. See Appendix Table S4 for the detailed comparison, P values, and sample number (n).

C  Clonogenic survival assay was performed to determine the sensitivity of NEO2734 in SPOP mutant C4-2 cells. The survival curve generated from three replicates ($n = 3$) showed IC50 for EV (1.01 μM), SPOP Q165P (0.621 μM) and SPOP F133V cells (0.395 μM).

D, E  C4-2 cells expressing EV, Q165, or F133V constructs were treated with 1 μM of NEO2734 for 4 days and cultured for another 8 days before harvesting. The number of colonies with more than 50 cells was counted. Representative images of colonies are shown in (D) with quantification data in (E). All data shown are means ± SEM, $n = 4$. The P value was calculated by the unpaired two-tailed Student's t-test; *$P < 0.05$, ***$P < 0.001$. See Appendix Table S4 for the detailed comparison, P values and sample number (n).

F–H  A schematic depicts the procedure of the establishment of SPOP mutant xenograft models and inhibitor administration (F). When the tumor reached 100 mm³, mice were treated with vehicle (40% polyethylene glycol), JQ1 (50 mg/kg) or NEO2734 (30 mg/kg) 5 days a week for three consecutive weeks. Tumors isolated from mice at day 21 of treatment were photographed (G) and tumor growth are shown in (H). All data shown are means ± SEM, $n = 6$. The P value comparing the tumor volume at day 21 post-treatment was calculated by the unpaired two-tailed Student's t-test; *$P < 0.05$, **$P < 0.01$; ***$P < 0.001$. See Appendix Table S4 for the detailed comparison, P values, and sample number (n).

Source data are available online for this figure.

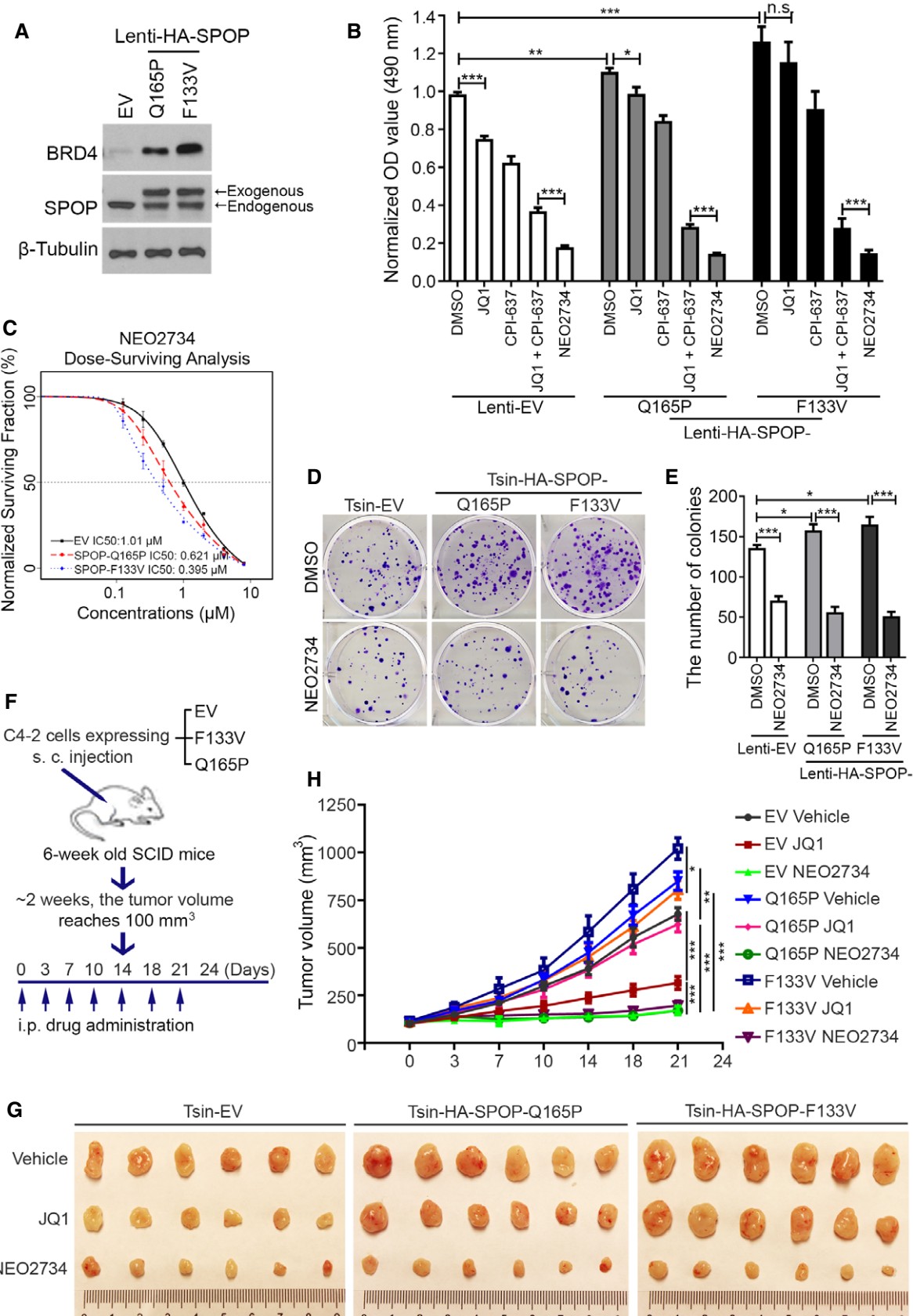

**Figure 7.**

# Materials and Methods

### Plasmids, chemicals, and antibodies

HA-tagged ubiquitin (HA-Ub) mammalian expression vector (#18712) was purchased from Addgene. Expression vectors for Myc-SPOP WT and F133V and Q165P mutants, Flag-SPOP WT and Q165P mutant, Myc-Cullin 3, Flag-BRD4 were generated or described previously (An *et al*, 2014; Zhang *et al*, 2017). SPOP mutant expression vectors were generated using KOD-Plus Muta-genesis Kit (#F0936K, Toyobo). MG132, CPI-637, JQ1 were purchased from Sigma-Aldrich. NEO2734 (also known as EP31670) is the result of a drug discovery collaboration between the University of Miami, Epigenetix Inc., and the Neomed Insti-tute. The activity of NEO2734 in inhibition of BET as well as p300/CBP bromodomains was initially confirmed with the BROMOscan platform (DiscoverX/Eurofins). Matrigel basement membrane Matrix (Cat No: 354248) was purchased from Corning Life Sciences.

The antibodies used are as follows: anti-Myc tag (9E10) (Cat No: sc-40; 1 in 1,000 dilution for Western blot (WB)) and Ac-H3 (Cat No: sc-56616; 2 µg for each chromatin immunoprecipitation (ChIP) reaction) from Santa Cruz Biotechnology; anti-HA (Cat No: MMS-101R; 1 in 1,000 dilution for WB) from Covance, Flag-M2 (Cat No: F-3165; 1 in 1,000 dilution for WB; 2 µg for each immunoprecipitation (IP) reaction), AR (H-280) (Cat No: sc-13062, 1 in 1,000 dilution for WB; 1 in 500 dilution for IF) from Sigma; β-tubulin (9F3) (Cat No: 2128S; 1 in 1,000 dilution for WB), AKT (Cat No: 9272S; 1 in 1,000 dilution for WB), phospho-AKT-Ser473 (Cat No: 9271S; 1 in 1,000 dilution for WB; 1 in 500 dilution for IF), phospho-S6K-Thr389 (Cat No: 9205S; 1 in 1,000 dilution for WB), S6K (Cat No: 9202S; 1 in 1,000 dilution for WB), cleaved caspase-3 (Cat No: 9579S, 1 in 500 dilution for IF), and PTEN (Cat No: 9559S; 1 in 1,000 dilution for immunohistochemistry (IHC)) from Cell Signaling Technology; AR (Cat No: ab108341; 1 in 10,000 dilution for IHC), BRD2 (Cat No: ab139696; 1 in 10,000 dilution for IHC; 1 in 1,000 dilution for WB), BRD4 (Cat No: ab128874; 1 in 10,000 dilution for IHC; 1 in 1,000 dilution for WB; 2 µg for each ChIP reaction), and Ki67 (Cat No: ab15580; 1 in 1,000 dilution for IF) from Abcam; BRD3 (Cat No: A302-368A; 1 in 1,000 dilution for both IHC and WB) from Bethyl Lab; SPOP (Cat No: 16750-1-AP; 1 in 1,000 dilution for WB) from Proteintech Group Inc; E-cadherin (Cat No: 610181; 1 in 1,000 dilution for WB; 1 in 500 dilution for IF), RAC-1 (Cat No: 610650; 1 in 1,000 dilution for WB) and SRC3 (Cat No: AIB-1, 611105; 1 in 1,000 dilution for WB) from BD Biosciences.

### Cell lines, cell culture, transfection, and lentivirus infection

DU145, LNCaP, C4-2, and 293T cells were obtained from the American Type Culture Collection (ATCC). The 293T cells were maintained in DMEM with 10% FBS, and DU145, LNCaP, and C4-2 cells were maintained in RPMI medium with 10% FBS. Cells were transiently transfected using Lipofectamine 3000 (Thermo Fisher Scientific) according to the manufacturer's instructions. For the establishment of stable cell lines, the pTsin-HA-SPOP mutant expression and virus packing constructs were transfected into 293T cells. Virus supernatant was collected 48 h after transfection.

DU145 or C4-2 cells were infected with viral supernatant in the presence of polybrene (8 µg/ml) and then selected in growth media containing 1.5 µg/ml puromycin. All cell lines used in this study have been recently authenticated by CellCheck 9 (9 Marker STR Profile and Inter-species Contamination Test) and tested for mycoplasma contamination by Lookout Mycoplasma PCR Detec-tion Kit (Cat No: MP0035; Sigma-Aldrich). Meanwhile, plasmocin (InvivoGen) was added to cell culture media to prevent myco-plasma contamination.

### PCa patient samples and SPOP mutation detection

All experiments involving the use of human samples conformed to the principles set out in the WMA Declaration of Helsinki and the Department of Health and Human Services Belmont Report. Forma-lin-fixed, paraffin-embedded (FFPE) samples from radical prostatec-tomy of 96 patients with clinically localized PCa were randomly selected from Mayo Clinic Tissue Registry. All cases upon collec-tion into the registry had been pathologically reviewed. The Mayo Clinic institutional review board (IRB) approved the experimental protocols for retrieving pathology blocks/slides, for accessing elec-tronic medical records, and experiments in this study. For Sanger sequencing, DNA was extracted from all 96 cases of FFPE PCa tissues using a QIAamp DNA FFPE Tissue Kit (Qiagen, Cat No. 56404), followed by PCR to amplify SPOP fragments containing exon 6 or 7. PCR products were purified using QIAquick Gel Extraction Kit (Qiagen, Cat No. 28704) according to the manufac-turer's instruction and used for Sanger sequencing. The primers used for DNA amplification are listed in Appendix Table S1. The forward primer used for amplifying exon 6 and the reverse primer used for exon 7 were also used for Sanger sequencing. Genomic DNA was extracted from PDX tumor tissues and organoid culture by using QIAamp Fast DNA Tissue Kit (QIAGEN; Cat No. 51404). The SPOP fragment containing the Q165 residue was amplified using Bio-Rad PCR machine. The primer sequences are listed in Appendix Table S1.

### Subrenal capsule grafting and development of transplantable tumor lines from patients

The SPOP Q165P mutant PDX (LTL573R or 573R) was generated from the biopsy of liver metastasis from a CRPC patient in the Living Tumor Laboratory (www.livingtumorlab.com). SPOP WT PDX (LTL313HR or 313HR) is a CRPC model developed in castrated mice from LTL313H, which was derived from hormone naïve primary PCa biopsy of a patient in the Living Tumor laboratory. Within 24 h of sample arrival, biopsy tissue was grafted into the subrenal capsule of male SCID mice using the method as described previously (Wang *et al*, 2005; Lin *et al*, 2014). Growing tumors (transplantable tumor lines) were consistently maintained by serial subrenal capsule transplantation. Xenografts were harvested, measured and fixed for histopathological analysis. BC Cancer Research Centre IRB approved the experimental protocols for obtaining the biopsy of liver metastasis and FFPE patient tumor blocks/slides, and experi-ments in this study. Animal care and experiments were carried out in accordance with the guidelines of Canadian Council on Animal Care and were approved by the Institutional Animal Care and Use Committee (IACUC).

## PDX maintenance and organoid culture

All experiments involving the use of human samples conformed to the principles set out in the WMA Declaration of Helsinki and the Department of Health and Human Services Belmont Report. LTL313HR (SPOP WT), LTL573R (SPOP Q165P), and V2-Met (SPOP WT) PDXs were transplanted subcutaneously and maintained in SCID male mice. When the tumors reached the size of 1,000 mm$^3$ about 2 months, they were passaged down to the next generation. The protocols for PDX experiments were approved by IACUC of Mayo Clinic. The more detailed information about these three PDXs is listed in Appendix Table S2.

PDX-derived organoid cultures were carried out using the protocol as described previously (Drost et al, 2016). Briefly, the tumor was cut into ∼1 mm$^3$ pieces and digested in 5 mg/ml collagenase with 10 μM Y-27632 for 1 h at 37°C. The digested tumor tissue was washed with adDMEM/F12 medium and then treated with TrypLE for 10 min at 37°C. The cell suspension containing 50,000 cells was spun down to remove TrypLE and resuspended with 40 μl of Matrigel, followed by plated onto 24-well plate. The cultured medium for all the organoids was freshly prepared as described previously (Drost et al, 2016). The more detailed information about these six organoids is listed in Appendix Table S3.

## Molecular dynamics simulation

The missing loops in the crystal structure of SPOP dimer (3HQI.pdb residue 24–331) were built by Modeller (Fiser et al, 2000). The above built structural model of the SPOP dimer was solvated in a cubic box of TIP3P water with a buffer of 10 Å. NaCl (0.15 M) was added to neutralize the system. The system consists of 201,262 atoms, including 63,714 water molecules, 616 residues from protein, 193 Na$^+$ and 174 Cl$^-$. The molecular dynamics simulations of WT and Q165P mutant were performed using AMBER 14 (Case et al, 2006). Particle mesh Ewald (PME; Darden et al, 1993) was applied to handle long-range electrostatic interactions with a cutoff of 12 Å. Van der Waals interactions were calculated with a cutoff of 12 Å. The CHARMM36 force field was used to compute the interactions within the protein (Hornak et al, 2006). The SHAKE algorithm (Ryckaert et al, 1977) was employed to constrain bonds including hydrogen atoms, thus allowing for a time step of 2 fs. The solvated systems were first minimized by 5,000 steps (2,500 steps of the steepest descent plus 2,500 steps of a conjugate gradient) with the backbone of protein constrained at force constant of 1.0 kcal/mol/Å$^2$, followed by initial equilibration of 25 ps at 303.15 K at NVT ensemble. Finally, the simulation was run for 80 ns at NPT ensemble for production without any constraints.

## ChIP-qPCR, RNA extraction, and reverse transcription–quantitative PCR (RT–qPCR)

ChIP-qPCR was performed as described previously (Boyer et al, 2005; Zhang et al, 2017). Briefly, after the treatment with DMSO, JQ1 (2 μM), CPI-637 (2 μM), the combination of JQ1 (2 μM) and CPI-637 (2 μM) or NEO2734 (2 μM), DU145 or C4-2 cells were harvested for ChIP with BRD4 or Ac-H3 antibodies. The DNAs pulled down by antibodies or non-specific IgG were amplified by qPCR. The values were normalized to the individual input. Total

RNAs were extracted with TRIzol (Invitrogen). Two micrograms of RNA was used for cDNA synthesis with SuperScript III First-Strand Synthesis System (Promega). Quantitative PCR (qPCR) was carried out in the iQ thermal cycler (Bio-Rad) using the iQ SYBR Green Supermix (Bio-Rad). Each sample was used in triplicate, and three biological repeats were performed. The ΔCT was calculated by normalizing the threshold difference of a certain gene with glyceraldehyde-3-phosphate dehydrogenase (GAPDH). The primer sequences are listed in Appendix Table S1.

## Protein extraction and Western blot analysis

Cultured cells were washed with PBS and lysed in cell lysis buffer (25 mM Tris–HCl pH 7.4, 150 mM NaCl, 1 mM EDTA, 1% NP-40, and 5% glycerol). Frozen PDX tissues were ground into powder on dry ice before adding the lysis buffer. Both cultured cells and ground tumor tissues were incubated for 30 min on ice and centrifuged at 15,600 g for 15 min to remove the debris before used for Western blot analysis.

## Hematoxylin and eosin staining (H&E), immunohistochemistry (IHC), and immunofluorescent cytochemistry (IFC)

Paraffin tissue sections for H&E, IHC, and IFC were cut at 4 μm thickness. H&E and IHC were performed according to a previously published study (Blee et al, 2018). Specifically, the antigen retrieval was conducted via heat-induced epitope retrieval in 10 mM sodium citrate buffer (pH 6.0) for all antibodies used in this study. Antibodies were diluted at appropriate concentrations as required and incubated in a humidified box overnight at 4°C. The tissue sections were further incubated with SignalStain®Boost IHC Detection Reagent (HRP, Rabbit), and the staining was developed with SignalStain®DAB Substrate Kit.

For IFC on PDX tissues, all the steps were the same as IHC paraffin tissues except that secondary fluorescence antibodies (Alexa Fluor 488 and Alexa Fluor 594) were used at 1 in 500 dilution. For IFC on the organoids, the organoid cultures were smeared onto slides and fixed with ice-cold methanol/acetone (1:1 dilution) at 4°C for 10 min, followed by three washes with 0.2% Triton X-100 in PBS. The organoids were blocked with 10% goat serum at room temperature for 1 h and followed by primary antibody incubation at 4°C overnight. In the following day, the organoids were incubated with secondary antibody at room temperature for 1 h. The nucleus was counterstained with DAPI.

## MTS cell proliferation assay and clonogenic survival assay

For MTS cell proliferation assay, cells were plated at a density of 2,000 cells/well in 96-well plates. At 4 h after plating, cells were treated with different concentrations of drugs and harvested at 72 h post-treatment. The OD value was read at a wavelength of 490 nm. The clonogenic survival assay was conducted as previously described (Yan et al, 2018). Briefly, an appropriate number of cells for different dosages of drugs were plated onto 6-well plate. At the following day, cells were treated with DMSO or NEO2734 for 4 days and then cultured with fresh medium without drugs for another 8 days. 12 days later, colonies were fixed and stained with crystal violet 0.5% (w/v) for 1 h. The colonies with more than 50 cells were counted, and the

## The paper explained

### Problem

Prostate cancer is one of the leading causes of cancer death in American and European men. SPOP-mutated prostate cancer possesses distinct molecular features such as resistance to BET inhibitors. There is an urgent need to define novel therapeutics for this unique subtype of prostate cancer in clinic.

### Results

There are a number of novel discoveries in the present study: (i) identification of a previously unrecognized Q165P mutation at the edge of SPOP MATH domain in primary prostate cancer and a liver metastasis; (ii) Q165P likely causes structural changes in the MATH domain and impairs the ability of SPOP in substrate binding, dimerization, and substrate degradation; (iii) bona fide SPOP substrates such as AR, SRC3 and BRD4 and AKT signaling were aberrantly upregulated in Q165P mutated PCa cells; (iv) Q165P mutant patient-derived xenografts (PDXs) and organoid models were established from patient samples; (v) Q165P mutant PCa cells were modestly responsive and hotspot mutant F133V cells were resistant to the BET inhibitor JQ1, but they were all highly sensitive to NEO2734, a novel dual inhibitor of BET and CBP/p300 *in vitro* and *in vivo*.

### Impact

SPOP-mutated subtype of prostate cancer bears several unique features such as aberrant AKT and AR co-activation and BET inhibitor resistance. Our findings obtained from novel SPOP-mutated prostate cancer PDX and organoid models stress that prostate cancers, especially SPOP-mutated subtype, can be effectively treated by the BET and CBP/p300 dual inhibitor.

number of colonies in drug-treated groups was normalized to the untreated group. The linear regression was applied to generate survival curves by R software (version 2.15.0; http://www.r-project.org).

### Drug treatment of PDX and xenograft tumors

The 6- to 8-week-old SCID male mice for both PDX and xenograft drug treatment experiments were generated in house and maintained in standard condition with a 12-h light/dark cycle and access to food and water *ad libitum*. The protocols for PDX and xenograft drug treatment experiments were approved by IACUC of Mayo Clinic. SPOP WT and Q165P mutant PDX tumors were established by passaging tumor pieces (~1 mm$^3$) subcutaneously (s.c.) into 6- to 8-week-old SCID male mice. After tumors reach ~100 mm$^3$ in size (approximately 4 weeks after transplantation), tumor-positive animals in both SPOP WT and Q165P groups were randomly divided into five treatment groups (at least five mice/group). JQ1, CPI-637, and NEO2734 were dissolved in 40% polyethylene glycol (PEG400). Mice were treated via intraperitoneal (IP) injection with the vehicle (40% PEG400), JQ1 (50 mg/kg), CPI-637 (30 mg/kg), JQ1 plus CPI-637 or NEO2734 (30 mg/kg). Xenograft tumors were generated using the protocol as described previously (Yan *et al*, 2018). Briefly, 5 × 10$^6$ C4-2 cells expressing EV, F133V, or Q165P constructs were injected into the left flank of SCID male mice. When the tumors reached 100 mm$^3$, mice were randomly divided into three groups (at least five mice/group) for the treatment with vehicle (40% PEG400), JQ1 (50 mg/kg) or NEO2734 (30 mg/kg). For both PDX and xenograft models, mice were treated with these drugs

5 days a week for 21 consecutive days. Tumor growth was measured blindly by caliper twice a week. The tumor volume was calculated using the formula 0.5 × Length (L) × Width (W)$^2$. When the first tumor reached a volume of 1,000 mm$^3$, the treatment was terminated and tumors were harvested for the photograph.

### Software usage, generation of graphs, and statistical analysis

ImageJ was downloaded from https://imagej.nih.gov/ij/download.html and used to analyze the bands of Western blot and E-cadherin intensity of IFC. Graphs were generated by using GraphPad Prism 8 project (GraphPad Software Inc, CA, USA) or R software version 2.15.0 (http://www.r-project.org). The original datasets for all the graphs are shown in Dataset EV1.

For PDX and C4-2 cell xenograft mouse studies, at least five age- and tumor size-matched male SCID mice were randomly grouped for the drug treatment and the tumor growth was monitored blindly. All results with error bars shown in this study were generated from at least three biological replicates. Data are shown as mean values ± SEM. The *P* value between the two groups was evaluated by the two-tailed student *t*-test or Wilcoxon rank-sum test with continuity correction as indicated. The *P* value among several treatment groups was evaluated by one-way ANOVA. The symbols for statistical analysis are as follows: n.s., not significant; \**P* < 0.05, \*\**P* < 0.01, \*\*\**P* < 0.001. The detailed *P* values and sample number (*n*) are shown in Appendix Table S4 for each experiment.

**Expanded View** for this article is available online.

## Acknowledgements

This work was supported in part by grants from the National Institutes of Health (CA134514, CA130908, CA193239, and CA203849 to H.H.), the Mayo Clinic Foundation (to H.H.), Epigene Therapeutics Inc (to H.H.), the Canadian Institutes of Health Research operating grants (141635, 144159, and 153081 to Y.W.), Terry Fox Research Institute program project (1062 to Y.W.), and the National Natural Science Foundation of China (81672544 and 81872099 to D.Y.). The authors would like to thank Epigene Therapeutics Inc for providing the compound NEO2734.

## Author contributions

HH conceived the study. YY, JM, DW, SW, YZ, LS, YP, CW, YC generated reagents and performed experiments, data collection, and analysis. XP performed computation simulation and modeling of SPOP structure. DL, HX, and YW generated PDXs. JZ, DL, MEG, CCC, YW, YY acquired patient tissue samples, performed IHC, and scored the data. HH, FJG, YW, DY, YY, JM, DW, DL wrote the manuscript.

## Conflict of interest

H.H. received research funding from Epigene Therapeutics Inc (Montreal, QC, Canada) and F.G. formerly served as a consultant to, and has equity interest in, Epigene Therapeutics Inc. C.W. is a co-founder of Epigenetix Inc. The authors declare that they otherwise have no conflict of interest.

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
