## [Review Process File · EMBO Molecular Medicine]

The novel BET-CBP/p300 dual inhibitor NEO2734 is active in SPOP mutant and wild-type prostate cancer

Yuqian Yan, Jian Ma, Dejie Wang, Dong Lin, Xiaodong Pang, Shangqian Wang, Yu Zhao, Lei Shi, Hui Xue, Yunqian Pan, Jun Zhang, Claes Wahlestedt, Francis J. Giles, Yu Chen, Martin E. Gleave, Collin C. Collins, Dingwei Ye, Yuzhuo Wang and Haojie Huang.

Review timeline:

Submission date:	25 th March 2019
Editorial Decision:	9 th May 2019
Revision received:	9 th August 2019
Editorial Decision:	17 August 2019
Revision received:	20 th August 2019
Accept:	4 th September 2019

Editor: Celine Carret

Transaction Report:

1st Editorial Decision

9th May 2019

Thank you for the submission of your manuscript to EMBO Molecular Medicine. I am sorry for the belated decision but in this case we experienced unusual difficulties in securing three willing and appropriate reviewers. Although I was hoping to obtain a third evaluation, this referee is very late so I am now proceeding based on the two consistent evaluations obtained so far as further delays cannot be justified.

You will see that in aggregate, both reviewers find the study of certain interest, while at the same time clearly mentioning current limitations that would need to be thoroughly addressed in a revised article. You will see that both referees are concerned about the novelty of the findings and their main issue relates to the choice of studying a rare SPOP Q165P mutation, clouding the clinical relevance for PCa. Still, we believe that referee #2 provides valuable suggestions to improve the paper on both fronts. In addition, better details, explanations, statistics, grammar/syntax usage and quantitative analysis should be provided, as well as a more focused discussion. I'd like to add that referee #1 further added the following after our cross-commenting exercise, which we would strongly encourage you to address: "Given [the mutant] is not very common as they showed, I won't think it is clinically relevant unless they provide additional following data: A. [...] use matched PDXs and PDOs, where one is wt and the other is point mutant Q165P and correlate it with AR levels CRPC. and B. [...] access AR IHC data from large cohort and correlate with this mutation."

If you feel you can satisfactorily address these points and those listed by the referees, you may wish to submit a revised version of your manuscript.

REFeree REPORTS

Referee #1 (Comments on Novelty/Model System for Author):

While biochemical experiments are robust, they rely often on ectopic expression of the mutant in question.

The same group identified a similar point mutation earlier in the same domain doing the same thing. So, not sure how novel is this.

It appears there aren't many PCa that express this mutant. It doesn't seem to cause resistance to BET inhibitors.

Referee #1 (Remarks for Author):

In this manuscript authors claim to have identified a novel loss-of-function Q165P mutation in tumour suppressor SPOP in a clinical specimen of primary PCa.

They show that this mutation impairs substrate binding and homo-dimerization abilities of SPOP leading to increased expression and activation of AR, bromodomain and SRC3, and activation of AKT, thus linking this to tumour growth activation and liver mets.

Using cellular and PDX systems they further show that Q165P mutant expressing cancer cell lines exhibit bromodomain inhibitor JQ1 resistance but can be sensitised by a combined blockage of BRD (using inhibitor JQ1) and CBP (using inhibitor CPi637). Using a dual inhibitor of BRD and CBP, the NEO2734 compound they claim to achieve enhanced sensitisation of Q165P to growth inhibition by NEO compound.

Overall, the study mirrors a previous study that identified a point mutation (F133V) with analogous phenotypic effects, In fact the earlier F133V appears to be more relevant to SPOP activity than Q165P both biologically and in clinical terms.

Specific comments:

1. What is the source of dual inhibitor NEO2734 compound?
2. In localised PCa, what % of tumours individually (or collectively) harbour two point mutations mentioned and studied in this work.
3. Page 6, last line- the word "largely" isn't used correctly, it is either "large" or "larger" more appropriately and should be corrected.
4. Fig 1D: it isn't clear why author did not do IHC for BRD4 in tumours, while they used BRD4 for biochemical studies in further figures. The qualification of in situ staining also appears to be missing.
5. Fig 2D: authors show increased expression (or activation) of some SPOP targets, they should use F133V as a positive control in this experiment.
6. Fig 2F/6D: I.F. data have not been quantified, it appears Q165P enhanced E-Cad level and will be predicted to hinder EMT. Given the authors's claim that Q165P is linked with tumour progression, it will be counter intuitive as the authors show the presence of homozygous Q165P in liver mets.
7. Fig. 3G: It does not appear that CULLIN3 binds weakly to Q165P mutant SPOP compared to WT SPOP, the W.B. image should be normalised and band intensity quantification should be provided to substantiate the claim.
8. Fig. 4B: the evidence that Q165P expression leads to JQ1 is weak. It is also unclear why dual inhibitor or BET and CBP, the NEO2734 compound shows greater inhibition than achieved by targeting both pathways individually.
9. Fig. 4D/H: clonogenic assays: In the absence of absolute colony numbers, it appears that EV produces more clones than Q165P which contradicts with the main message of this work. If this is indeed a case, what is the likely explanation?

Referee #2 (Comments on Novelty/Model System for Author):

See details in the author comments below. Technical quality suffers from a lack of detail on numbers of experiments and statistical data with many representative images used, although the actual models used are excellent. While the novelty of the mutation is high, the medical impact would be substantially improved by addition of other SPOP mutations, which are more common than the mutation examined. It may be better presented as a short report to reflect the more narrow implications.

Referee #2 (Remarks for Author):

This study follows on from a recent Nature Medicine publication examining SPOP mutation in prostate cancer. The authors characterise a novel SPOP mutation, generating data on how it affects protein interactions, and generate novel models including organoids and PDXs from this patient sample. While the data are generally interesting, this is not a hotspot mutation, which questions whether their data is generalizable to other SPOP mutations. Further analysis of the inhibitors in other SPOP mutant organoids would greatly benefit the manuscript and strengthen their conclusions.

General comments and questions:

1 - There are many typographical errors in the text and figures, which should be addressed. Examples include page 6 "largely" should be "large", Figure 1C "trancating" should be "truncating", but there are many errors and grammatical changes that would assist the reader in interpreting the data.

2 - The data presented in Figure 1D should be expanded to more samples, as selection of a single SPOP WT for comparison is not compelling. Analysis of BRD3, and other BRD family members in additional WT samples and SPOP mutant samples would assist in determining whether this expression difference is due to SPOP mutation.

3 - The number of samples in the "large" prostate cancer cohort in Figure 1 should be stated in the legend.

4 - In general, figure legends lack appropriate acknowledgment of the number of repeat experiments performed, and this should be added throughout the manuscript to allow the reader to interpret the robustness and significance of the data.

5 - T1 and T2 should be explained in the legend of Figure 2D.

6 - Figure 3 analysis should be done for the F133V mutant SPOP as a positive control to show that the modelling works, and compare the changes seen between the two mutants. Indeed, it would be interesting to undertake the modelling of a range of the hotspot mutations, to determine whether these mutations are all predicted to alter the MATH domain structure, particularly those present in the various organoid models available.

7 - Figure 4D and 4H require further quantitative analysis of colony number and size, with associated statistics.

8 - Additional analysis of the organoids treated in Figure 5G would be helpful. Is it possible to look for markers of proliferation or apoptosis, rather than only using organoid size as a marker of inhibition?

9 - How was the control SPOP wt PDX model chosen? Analysis of other SPOP wt models by western blot, IHC and IFC in Figure 6B, C and D would be helpful in determining whether these effects are due to SPOP mutation or just differences in these two PDX models. Analysis of pAKT by IFC in Figure 6D would also be beneficial.

10 - While it is clear and consistent in this study that the dual inhibition is superior to JQ1 alone, the author statements around both BET inhibitor resistance of SPOP as well as the specificity of the dual inhibition for SPOP mutant are somewhat overstated. The previous Nature Medicine publication clearly showed for the F133V mutant that it was refractory to JQ1, which the WT was sensitive. However in this current study, both WT and SPOP mutant cells respond to the JQ1 inhibitor, and in both, the dual inhibition increases the response (Figure 7B-C). The same holds true for AR, pAKT and pS6K, all of which are reduced in both WT and mutant PDXs. While it is true that JQ1 has a smaller inhibitory effect in mutant PDXs, this is not consistent with BET inhibitor "resistance" as was shown for the F133V mutant in the Nature Medicine manuscript. The authors should tone down these claims to more accurately reflect their data, which suggest that dual inhibition works well in both SPOP wt and mutant cells. Discussion around the differences between these two mutants should also be incorporated. In order to show that the inhibitors overcome the BET inhibitor resistance, using the F133V mutant model would be required, or perhaps one of the organoid models with SPOP mutants described in the discussion.

1st Revision - authors' response

9th August 2019

Authors' Response to Editor's and Reviewers' Comments on the Manuscript EMM-2019-10659

Editor's Comments:

You will see that both referees are concerned about the novelty of the findings and their main issue relates to the choice of studying a rare SPOP Q165P mutation, clouding the clinical relevance for PCa. Still, we believe that referee #2 provides valuable suggestions to improve the paper on both fronts. In addition, better details, explanations, statistics, grammar/syntax usage and quantitative analysis should be provided, as well as a more focused discussion.

Reply: To increase the clinical relevance of the current study to PCa, we have examined the anti-cancer efficacy of the novel compound NEO2734 on the other SPOP mutations (e.g. hotspot mutations F133V and W131R) using both organoids derived from PCa patients and PDX tumors. We also performed new xenograft studies by testing the anti-cancer efficacy of this novel compound on the SPOP hotspot mutant F133V *in vivo*.

Additionally, as suggested by the Editor and Reviewers, we have provided better details, explanations, statistics, grammar/syntax usage and quantitative analysis as well as a more focused discussion.

I'd like to add that referee #1 further added the following after our cross-commenting exercise, which we would strongly encourage you to address: "Given [the mutant] is not very common as they showed, I won't think it is clinically relevant unless they provide additional following data: A. [...] use matched PDXs and PDOs, where one is wt and the other is point mutant Q165P and correlate it with AR levels CRPC. and B. [...] access AR IHC data from large cohort and correlate with this mutation."

Reply: We thank the Editor and the Reviewer #1 for excellent suggestion. To address the point "A", we analyzed AR protein expression by performing AR IHC staining in matched WT and Q165P PDX tissues along with a cohort of 20 CRPC patient samples. Similar to the western blot data shown in Figure 2D, AR expression was higher in Q165P PDX tissue compared to that in WT counterpart PDX (Fig EV1D and E). More importantly, AR protein level was slightly higher than the average AR protein level in CRPC tissues (Fig EV1E). Additionally, we also employed both western blot and IFC approaches to examine AR protein expression in the PDO derived from matched WT (313HR) and Q165P (573R) PDX tumors and four more different patient-derived PCa organoids as reported previously (Gao et al., Cell 159: 176-187, 2014; Zhang et al., Nature Medicine 23: 1055-1062, 2017). Both methods showed that similar to the results in PDX tumors (Figure 2D, Fig EV1E, EV3B and D), AR protein level was higher in the organoids from Q165P PDX (573R) than the matched WT PDX (313HR) (Fig 6A and EV5E). It is worth noting that AR expression level was also very high in two other SPOP WT organoids (Fig 6A, Fig EV5E), indicating that there may

be SPOP-independent mechanism that regulates AR protein expression in those two types of organoids.

To address the point “B”, we performed IHC to examine AR protein expression in Q165P patient tissue along with a large cohort of primary PCa patient samples (85 SPOP WT cases and 11 SPOP mutated cases). The new data showed that similar to the results in other SPOP mutated samples, AR protein expression was higher in Q165P patient sample compared to SPOP WT patient samples (Fig 1F and G).

Referee #1 (Comments on Novelty/Model System for Author):

While biochemical experiments are robust, they rely often on ectopic expression of the mutant in question. The same group identified a similar point mutation earlier in the same domain doing the same thing. So, not sure how novel is this. It appears there aren't many PCa that express this mutant. It doesn't seem to cause resistance to BET inhibitors.

Referee #1 (Remarks for Author):

In this manuscript authors claim to have identified a novel loss-of-function Q165P mutation in tumour suppressor SPOP in a clinical specimen of primary PCa. They show that this mutation impairs substrate binding and homo-dimerization abilities of SPOP leading to increased expression and activation of AR, bromodomain and SRC3, and activation of AKT, thus linking this to tumour growth activation and liver mets. Using cellular and PDX systems they further show that Q165P mutant expressing cancer cell lines exhibit bromodomain inhibitor JQ1 resistance but can be sensitized by a combined blockage of BRD (using inhibitor JQ1) and CBP (using inhibitor CPi637). Using a dual inhibitor of BRD and CBP, the NEO2734 compound they claim to achieve enhanced sensitization of Q165P to growth inhibition by NEO compound. Overall, the study mirrors a previous study that identified a point mutation (F133V) with analogous phenotypic effects. In fact the earlier F133V appears to be more relevant to SPOP activity than Q165P both biologically and in clinical terms.

Specific comments:

1. What is the source of dual inhibitor NEO2734 compound?

Reply: NEO2734 (also known as EP31670) is the result of a drug discovery collaboration between the University of Miami, Epigenetix Inc. (Delray Beach, FL) and the Neomed Institute (Saint-Laurent, Canada). Activity of NEO2734 in inhibition of BET as well as p300/CBP bromodomains was initially confirmed with the BROMOScan platform (DiscoverX/Eurofins). This new information has been included in the “Plasmids, chemicals and antibodies” section in Materials and Methods in the revised manuscript.

2. In localised PCa, what % of tumours individually (or collectively) harbour two point mutations mentioned and studied in this work.

Reply: We have added in the revised manuscript the information about the incidence rate of the homozygous mutation. We indicated that exome sequencing was carried out in a cohort of 22 patients with PCa (Collins CC and Wang YZ, unpublished data). Among the tumors from these patients analyzed by sequencing, the Q165P mutation was heterozygous in the primary tumor and homozygous in the liver metastasis. Therefore, no primary tumors we analyzed harbored two point mutations at Q165 residue, suggesting that the homozygous Q165P mutation might affect therapy response and tumor progression. That is indeed the main purpose of the current study to utilize the Q165P PDX model we generated to examine the response of Q165P mutant prostate cancer cells to different pathway inhibitors in vitro and in vivo.

3. Page 6, last line- the word "largely" isn't used correctly, it is either "large" or "larger" more appropriately and should be corrected.

Reply: We apologize for the typo and the error has been corrected.

4. Fig 1D: it isn't clear why author did not do IHC for BRD4 in tumours, while they used BRD4 for biochemical studies in further figures. The qualification of in situ staining also appears to be missing.

Reply: We agree with the Reviewer that it is important to perform BRD4 IHC staining in Fig 1D. We have presented BRD4 IHC data in Fig 1D in the revised manuscript and moved BRD3 IHC data to Fig EV1C. Also, as suggested by the Reviewer, we have provided the quantification data of BRD4 IHC staining in Fig 1E.

5. Fig 2D: authors show increased expression (or activation) of some SPOP targets, they should use F133V as a positive control in this experiment.

Reply: We agree with the Reviewer. We have repeated western blot experiments for Figure 2D by including samples from EV and F133V C4-2 xenograft tumors (see more details in Fig 7G) as negative and positive controls, respectively.

6. Fig 2F/6D: I.F. data have not been quantified, it appears Q165P enhanced E-Cad level and will be predicted to hinder EMT. Given the authors's claim that Q165P is linked with tumour progression, it will be counter intuitive as the authors show the presence of homozygous Q165P in liver mets.

Reply: This is an excellent point. We agree that there were quite few clustered spots with stronger E-Cadherin staining signals in both Q165P PDO and PDX samples (Fig 2F/6D in original submission) in comparison to those in WT counterparts. Intriguingly, we also noticed that unlike WT organoids or PDX samples, which display regular and intact shape of E-cadherin staining in almost all cells, most cells in Q165P mutant PDO and PDX samples have irregular pattern of E-cadherin staining (Fig 2F/6D in original submission, Fig 6B and Fig EV3D in the current version). Additionally, we found that a few Q165P cells even lost E-cadherin staining. To more accurately evaluate E-Cadherin protein level in both WT and Q165P samples, we have retaken the images from different PDX tumors and quantified E-Cadherin IFC OD value per cell area. Our new data showed that the overall fluorescence intensity of E-Cadherin was not significantly altered (Fig EV4A and B).

Furthermore, we examined E-Cadherin protein level in WT and Q165P PDX tumors using western blot and quantified the ratio of E-Cadherin versus β -Tubulin by ImageJ. The new data have shown that E-Cadherin protein level was comparable in WT and Q165P PDX tumors (Fig EV4C and D), suggesting that Q165P mutation has no significant effect on E-Cadherin protein level in PCa.

7. Fig. 3G: It does not appear that CULLIN3 binds weakly to Q165P mutant SPOP compared to WT SPOP, the W.B. image should be normalised and band intensity quantification should be provided to substantiate the claim.

Reply: We agree with the Reviewer. We have used the ImageJ software to quantify the W.B. images and the intensity of the bands of immunoprecipitated CULLIN3 was sequentially normalized to that of CULLIN3 input. The normalized values are shown in the revised Fig 3G.

8. Fig. 4B: the evidence that Q165P expression leads to JQ1 is weak. It is also unclear why dual inhibitor or BET and CBP, the NEO2734 compound shows greater inhibition than achieved by targeting both pathways individually.

Reply: This is an excellent point. The data shown in Fig 4B was obtained from DU145 PCa cell line. We repeated the same experiment in a different cell line (C4-2) by including the SPOP hotspot mutant F133V as a positive control. Similar to the results in DU145 cells, Q165P C4-2 cells were still responsive to JQ1 although the sensitivity was not as robust as in EV control cells (Fig 7B). However, consistent with our previous report in C4-2 cells (Zhang et al., *Nature Medicine* 23:1055-1062, 2017), F133V C4-2 cells were resistant to JQ1 (Fig 7B). Importantly, these data are supported by the western blot data showing that BRD4 protein was much higher in F133V C4-2 cells than that in Q165P C4-2 cells (Fig 7A). Thus,

different from the effect of hotspot mutant F133V, Q165P mutant PCa cells still modestly respond to JQ1, and this is probably because this mutant fails to elevate BRD4 protein expression to the level induced by F133V.

To address why dual inhibitor NEO2734 shows greater inhibition than that achieved by targeting both pathways individually, we examined BRD4 occupancy and histone H3 acetylation (Ac-H3) level in BRD4 known target genes (*c-MYC* and *RAC1*) in both DU145 and C4-2 cells using ChIP-qPCR approach. While NEO2734 and dual inhibition with JQ1 and CPI-637 achieved similar effect on BRD4 binding at *c-MYC* and *RAC1* gene loci in both cell lines (Fig 4F and EV2A), NEO2734 had greater inhibitory effect on Ac-H3 level at these two gene loci than the dual inhibition with JQ1 and CPI-637 (Fig 4G and EV2B). These data indicate that in comparison to the effect of the combination of CBP/p300 and BET inhibitors the greater inhibition achieved by the dual inhibitor NEO2734 is likely mediated by its greater effect on inhibition of Ac-H3 level.

9. Fig. 4D/H: clonogenic assays: In the absence of absolute colony numbers, it appears that EV produces more clones than Q165P which contradicts with the main message of this work. If this is indeed a case, what is the likely explanation?

Reply: We apologize for not being able to clearly label Fig 4D/H. As we mentioned in the Figure legend in the previous submission, original Fig 4D/H show the images of clonogenic assays in EV and Q165P DU145 cells (Panel D) or EV and F133V DU145 cells (Panel H) which were all treated with NEO2374.

To avoid confusion, we have repeated the clonogenic assays and included both mock (DMSO)- or NEO2734-treated EV and Q165P DU145 cells in new Figure 4D and the results of EV and F133V C4-2 cells in Fig 7D. Also, as suggested by the Reviewer, we have quantified the absolute colony numbers in new Fig 4E and Fig 7E. Indeed, the quantified data clearly show that without drug treatment, SPOP mutant cells (Q165P and F133V) produced more colonies than EV cells, which is consistent with the tumor growth-promoting role of SPOP mutations as demonstrated previously (Zhang et al., Nature Medicine 23: 1055-1062, 2017). In contrast, after NEO2734 treatment Q165P or F133V cells did produce fewer colonies than EV cells (New Fig 4D/4E and Fig 7D/E). These data are consistent with the main message/finding of our current work that the dual inhibitor NEO2734 is active in inhibition of growth of both SPOP mutant and wild-type PCa cells while our data also suggest that SPOP mutant cells are more sensitive to NEO2734 compared to SPOP wild-type cells.

Referee #2 (Comments on Novelty/Model System for Author):

See details in the author comments below. Technical quality suffers from a lack of detail on numbers of experiments and statistical data with many representative images used, although the actual models used are excellent. While the novelty of the mutation is high, the medical impact would be substantially improved by addition of other SPOP mutations, which are more common than the mutation examined. It may be better presented as a short report to reflect the more narrow implications.

Referee #2 (Remarks for Author):

This study follows on from a recent Nature Medicine publication examining SPOP mutation in prostate cancer. The authors characterise a novel SPOP mutation, generating data on how it affects protein interactions, and generate novel models including organoids and PDXs from this patient sample. While the data are generally interesting, this is not a hotspot mutation, which questions whether their data is generalizable to other SPOP mutations. Further analysis of the inhibitors in other SPOP mutant organoids would greatly benefit the manuscript and strengthen their conclusions.

Reply: This is an excellent point. To investigate whether our data is generalizable to other SPOP mutations, we have performed several new experiments by using cell lines expressing different SPOP mutants, organoids harboring different SPOP mutants, and SPOP mutant

patient tissue samples. Firstly, we have generated stably-expressing Q165P C4-2 cells and compared the effect of Q165P on SPOP downstream targets to the effect of other SPOP mutations (F102C and F133V, two hotspot mutations) (Fig 2C). Secondly, we have compared the alteration of AR and AKT signaling pathways in a total of six organoid lines, two of which harbor SPOP mutations (Q165P and W131R) and four others express wild-type SPOP (Fig 6A and B, Fig EV5E). These four additional organoid lines were originally generated in the laboratory of Dr. Yu Chen at Memorial Sloan Kettering Cancer Institute and have been reported previously (Gao et al., *Cell* 159: 176-187, 2014; Zhang et al., *Nature Medicine* 23: 1055-1062, 2017). The new data regarding the expression of SPOP-regulated signaling pathway genes in these organoids are shown in Fig 6 A and B, Fig EV5E. Thirdly, we also examined the drug sensitivity of these organoids with WT SPOP or different mutants of SPOP. The new data regarding how they responded to single inhibitor of CBP/p300 (CPI-637) or BET (JQ1) or the dual inhibitor NEO2734 are shown in Fig 6C and D.

In addition, to further strengthen the clinical relevance of the current study as this point is also raised by the other Reviewer, we compared the expression of BRD4 and AR proteins in Q165P mutated patient samples to the level of these proteins in a cohort of primary PCa and a group of CRPC patient samples. Our new data show that BRD4 and AR protein levels in Q165P mutant patient sample are comparable to those in patient samples harboring other SPOP mutants (Fig 1D-G). Furthermore, AR protein level in Q165P patient samples was slightly higher than the average AR protein level in CRPC patient tissues (Fig EV1E). Taken together, our new data suggest that the findings with the Q165P mutant are generalizable to other SPOP mutations.

General comments and questions:

1 - There are many typographical errors in the text and figures, which should be addressed. Examples include page 6 "largely" should be "large", Figure 1C "trancating" should be "truncating", but there are many errors and grammatical changes that would assist the reader in interpreting the data.

Reply: We thank the Reviewer for pointing these out. We have carefully read through our manuscript and corrected the typographical and grammatical errors in the text and Figures.

2 - The data presented in Figure 1D should be expanded to more samples, as selection of a single SPOP WT for comparison is not compelling. Analysis of BRD3, and other BRD family members in additional WT samples and SPOP mutant samples would assist in determining whether this expression difference is due to SPOP mutation.

Reply: This is an excellent point. We have analyzed BRD4 expression by expanding IHC staining in a cohort of 96 primary prostate cancer patient specimens, which include 85 SPOP WT and 11 SPOP mutated samples (Fig 1D). Quantitative analysis of BRD4 IHC data shows that similar to the majority of SPOP mutated samples, Q165P mutated prostate cancer (the red dot in Fig 1E) expressed much higher level of BRD4 protein than the majority of SPOP WT samples (Fig 1E). The BRD2 and BRD3 IHC staining results from WT and Q165P mutant patient samples are shown in Fig EV1C.

3 - The number of samples in the "large" prostate cancer cohort in Figure 1 should be stated in the legend.

Reply: We have stated the exact number in the legend of Fig 1C.

4 - In general, figure legends lack appropriate acknowledgment of the number of repeat experiments performed, and this should be added throughout the manuscript to allow the reader to interpret the robustness and significance of the data.

Reply: We thank the Reviewer for pointing this out. We have added the number of biological replicates (n) in the experiments we performed and provided detailed statistical analysis information in the figure legends.

5 - T1 and T2 should be explained in the legend of Figure 2D.

Reply: That is a good point. We have provided the explanation for “T1” and “T2” in the legend of Fig 2D.

6 - Figure 3 analysis should be done for the F133V mutant SPOP as a positive control to show that the modelling works, and compare the changes seen between the two mutants. Indeed, it would be interesting to undertake the modelling of a range of the hotspot mutations, to determine whether these mutations are all predicted to alter the MATH domain structure, particularly those present in the various organoid models available.

Reply: This is an excellent point. We have previously performed the computer simulation/modelling for F133V mutant using ERG as a SPOP substrate (An et al., *Molecular Cell* 59: 904-16, 2015). Consistent with the finding that F133V is a hotspot mutation of the F133 residue that is located in the substrate binding surface of the MATH domain of SPOP (Barbieri et al., *Nature Genet* 44: 685-689, 2012), our modeling data shows that the binding of F133V to the substrate (e.g. ERG) is disrupted (An et al., *Molecular Cell* 59: 904-16, 2015). Slightly different from F133V, the binding of Q165P to its substrates such as BRD4 was substantially reduced, but not completely abolished (Fig 2A). Consistent with the protein binding data, F133 resides within the MATH domain, but Q165 resides in the peripheral region of the MATH domain and adjacent to the BTB domain, which is responsible for dimerization of SPOP (Fig 1C). These findings are in agreement with our computer simulation data showing that Q165P mutation leads to a large conformational change in the MATH domain of SPOP, which might therefore subsequently affect the stability of SPOP to dimer and ultimately impair SPOP binding and ubiquitination of its substrates. We have added the new discussions in the Discussion section.

7 - Figure 4D and 4H require further quantitative analysis of colony number and size, with associated statistics.

Reply: We apologize for not including the quantification data for Fig 4D and 4H. As discussed above when we address the other reviewer’s question, we have repeated the clonogenic assays and included mock (DMSO)-treated EV and SPOP mutated cells. The new data from EV and Q165P DU145 cells treated with DMSO or NEO2734 are shown in Fig 4D and the results from EV and F133V C4-2 cells treated with DMSO or NEO2734 are in Fig 7D. We also quantified the absolute colony numbers in these experiments and the new data are shown in Fig 4E and Fig 7E.

8 - Additional analysis of the organoids treated in Figure 5G would be helpful. Is it possible to look for markers of proliferation or apoptosis, rather than only using organoid size as a marker of inhibition?

Reply: We thank the Reviewer for the great suggestion. We have performed IFC to examine the expression of the proliferation marker Ki-67 and apoptotic marker cleaved Caspase-3 in the organoids after the treatment with JQ1, CPI-637, the combination of JQ1 and CPI-637 or NEO2734. The new staining and quantified data are shown in Fig 6E and 6F, EV5F and 5G.

9 - How was the control SPOP wt PDX model chosen? Analysis of other SPOP wt models by western blot, IHC and IFC in Figure 6B, C and D would be helpful in determining whether these effects are due to SPOP mutation or just differences in these two PDX models. Analysis of pAKT by IFC in Figure 6D would also be beneficial.

Reply: This is an excellent point. The reason that we chose this SPOP wt PDX model is because this WT PDX is also PTEN-null and it matches with the PTEN loss background of Q165P PDX. The PTEN expression IHC data are shown in Fig 2E.

As suggested by the Reviewer, we have included another WT PDX sample in new experiments (Fig EV3B, 3C and 3D) and added IFC staining of p-AKT-S473 in Fig EV3D.

10 - While it is clear and consistent in this study that the dual inhibition is superior to JQ1 alone, the author statements around both BET inhibitor resistance of SPOP as well as the specificity of the dual inhibition for SPOP mutant are somewhat overstated. The previous Nature Medicine publication clearly showed for the F133V mutant that it was refractory to JQ1, which the WT was sensitive. However in this current study, both WT and SPOP mutant cells respond to the JQ1 inhibitor, and in both, the dual inhibition increases the response (Figure 7B-C). The same holds true for AR, pAKT and pS6K, all of which are reduced in both WT and mutant PDXs. While it is true that JQ1 has a smaller inhibitory effect in mutant PDXs, this is not consistent with BET inhibitor "resistance" as was shown for the F133V mutant in the Nature Medicine manuscript. The authors should tone down these claims to more accurately reflect their data, which suggest that dual inhibition works well in both SPOP wt and mutant cells. Discussion around the differences between these two mutants should also be incorporated. In order to show that the inhibitors overcome the BET inhibitor resistance, using the F133V mutant model would be required, or perhaps one of the organoid models with SPOP mutants described in the discussion.

Reply: That is an excellent point. We have performed several new experiments to experimentally address this concern. Specifically, we performed new mouse experiments using F133V-expressing C4-2 xenografts in a manner similar to what we did for the Nature Medicine publication (Zhang et al., *Nature Medicine* 23: 1055-1062, 2017), but treating mice with either JQ1 or the dual inhibitor NEO2734. Consistent with our previous report in the Nature Medicine paper, our new results demonstrated that C4-2 SPOP F133V tumors were resistant to JQ1 (Fig 7F-H). Importantly, the differences in the sensitivity of F133V and Q165P mutant tumors to JQ1 is consistent with the western blot data that F133V C4-2 tumors expressed higher protein level of BRD4, RAC1 and phosphorylated AKT than Q165P PDX (Fig 2D) and these factors are important for JQ1 resistance in F133V tumors (Zhang et al., *Nature Medicine* 23: 1055-1062, 2017). Notably, JQ1 resistance was overcome by the dual inhibitor NEO2734 in both F133V and Q165P C4-2 xenografts (Fig 7G and H). Thus, to more accurately reflect their data, we have toned down our claims by indicating in the revised manuscript that Q165P cells are modestly responsive to JQ1 in vitro and in vivo whereas the SPOP hotspot mutants such as F133V are resistant to JQ1. Importantly, we demonstrate that both SPOP hotspot mutants and Q165P mutant prostate cancer cells are sensitive to the BET and CBP/p300 dual inhibitor NEO2734 in vitro and in vivo.

Additionally, we also performed new experiments using prostate cancer organoid harboring the SPOP hotspot mutant W131R (Fig 6C and D). Similar to F133V-expressing C4-2 cells in culture and xenografts in mice, SPOP W131R mutant organoids were resistant to JQ1, but this resistance was overcome by the dual inhibitor NEO2734 (Fig 6C and D), thereby providing further support to our conclusion.

2nd Editorial Decision

17 August 2019

Thank you for the submission of your revised manuscript to EMBO Molecular Medicine. We have now received the enclosed reports from the referees that were asked to re-assess it. As you will see the reviewers are now globally supportive and I am pleased to inform you that we will be able to accept your manuscript pending final editorial amendments.

 REFEREE REPORTS

Referee #1 (Comments on Novelty/Model System for Author):

This is a much improved version and the authors have sufficiently addressed the comments raised by this reviewer.

Referee #2 (Comments on Novelty/Model System for Author):

The authors have added numerous new experiments that have adequately addressed the issues with

the model systems being used.

Referee #2 (Remarks for Author):

The authors have adequately addressed the questions from my original review. The new data strengthen their conclusions and widen the impact of their findings.

2nd Revision - authors' response

20th August 2019

Authors' Response to Editor's and Reviewers' Comments on the Manuscript EMM-2019-10659-V2

***** Reviewer's comments *****

Referee #1 (Comments on Novelty/Model System for Author):

This is a much improved version and the authors have sufficiently addressed the comments raised by this reviewer.

Reply: We thank the Reviewer for the positive comments.

Referee #2 (Comments on Novelty/Model System for Author):

The authors have added numerous new experiments that have adequately addressed the issues with the model systems being used.

Referee #2 (Remarks for Author):

The authors have adequately addressed the questions from my original review. The new data strengthen their conclusions and widen the impact of their findings.

Reply: We thank the Reviewer for the positive comments.

Corresponding Author Name: Haojie Huang, Yuzhuo Wang

Manuscript Number: EMM-2019-10659